# Smoothness Errors in Dynamics Models and How to Avoid Them

**Edward Berman** [* 1] **Luisa Li** [* 1] **Jung Yeon Park** [1] **Robin Walters** [1]

## Abstract

Modern neural networks have shown promise for solving partial differential equations over surfaces, often by discretizing the surface as a mesh and learning with a mesh-aware graph neural network. However, graph neural networks suffer from oversmoothing, where a node's features become increasingly similar to those of its neighbors. Unitary graph convolutions, which are mathematically constrained to preserve smoothness, have been proposed to address this issue. Despite this, in many physical systems, such as diffusion processes, smoothness naturally increases and unitarity may be overconstraining. In this paper, we systematically study the smoothing effects of different GNNs for dynamics modeling and prove that unitary convolutions hurt performance for such tasks. We propose relaxed unitary convolutions that balance smoothness preservation with the natural smoothing required for physical systems. We also generalize unitary and relaxed unitary convolutions from graphs to meshes. In experiments on PDEs such as the heat and wave equations over complex meshes and on weather forecasting, we find that our method outperforms several strong baselines, including mesh-aware transformers and equivariant neural networks. Our code is available at github.com/EdwardBerman/rayleigh_analysis.

## 1. Introduction

Solving partial differential equations (PDEs) is crucial across many scientific and engineering domains, including acoustics, fluid dynamics, and electrodynamics. Recently, neural networks have been explored as alternatives to analytic and traditional numerical methods for solving PDEs. Neural networks offer faster inference (Cui et al., 2024), discretization free solutions (Li et al., 2021), better robustness

---
[*]Equal contribution [1]Geometric Learning Lab, Northeastern University. Correspondence to: Edward Berman <eddieberman@g.harvard.edu>.

*Proceedings of the 43rd International Conference on Machine Learning*, Seoul, South Korea. PMLR 306, 2026. Copyright 2026 by the author(s).

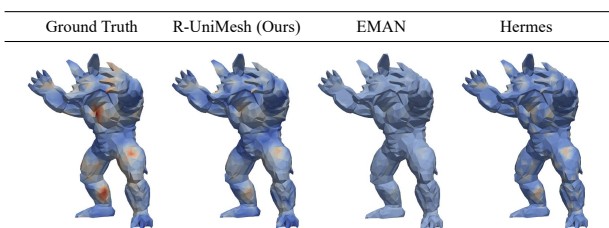

| Ground Truth | R-UniMesh (Ours) | EMAN | Hermes |

*Figure 1.* Qualitative comparison of autoregressive model predictions for the heat equation on the armadillo mesh at timestep $T = 190$. Our R-UNIMESH model remains faithful to the ground truth during each step of the rollout, whereas the EMAN model over smooths and the Hermes model under smooths. A more complete comparison over several timesteps is in Sec. C.4, Tab. 3.

to partial observability (Schlaginhaufen et al., 2021; Huang et al., 2024; Morel et al., 2025), and synergy with existing finite element methods (Gupta & Lermusiaux, 2023).

However, neural network models often have architectural biases that hurt their ability to model certain dynamics. In particular, many deep learning methods solve PDEs by discretizing the domain into a grid or mesh and modeling the solution using a graph neural network (e.g., Janny et al., 2023; Park et al., 2023). Unfortunately, graph neural networks (GNNs) tend to oversmooth (Li et al., 2018), where adjacent node features become increasingly similar over successive iterations of message passing. The phenomenon of oversmoothing occurs in a variety of settings (Cai & Wang, 2020; Bodnar et al., 2022; Keriven, 2022; Rusch et al., 2023; Kiani et al., 2024; Arroyo et al., 2025; Su & Wu, 2025) and hampers the performance of deep GNNs.

To address oversmoothing, Kiani et al. (2024) propose using unitary graph convolutions, which constrain weight matrices to be unitary. This ensures that the linear transformations preserve norms and remain invertible, improving network stability. They also prove that unitary convolutions prevent oversmoothing by preserving the Rayleigh quotient, a measure of smoothness for signals defined on graphs. However, this poses a new problem: many dynamics problems commonly solved using GNNs require *some* amount of smoothing. For example, heat diffusion on graphs and meshes naturally smooths the input node features. Using unitary graph convolutions in such problems would result in undersmoothing and does not give a complete solution to

smoothness errors.

In this work, we first theoretically characterize the limitations of unitary convolutions for dynamics problems. In particular, we derive a lower bound on the approximation error of unitary functions and show that unitary functions are overconstrained for dynamical systems where the solution's norm has high angular dependence. To address this issue, we propose *relaxed unitary convolutions*, which balance smoothness preservation with modeling fidelity, outperforming existing methods on dynamic systems that require natural smoothing. We also generalize both the Rayleigh quotient and unitary convolution framework from graphs to meshes so that relaxations can be applied in this setting. Finally, we systematically investigate smoothness tendencies of different mesh-GNN architectures and find that our mechanism for approximately preserving smoothness is key to successful modeling, providing equal or greater improvement to other inductive biases such as equivariance.

In summary, our contributions are the following:

1. Derive a lower bound on the approximation error of unitary functions, demonstrating that they are overly restrictive when predicting dynamics with high angular dependence in the solution's norm (Sec. 4).

2. Introduce relaxed unitary convolutions that balance accuracy with smoothness preservation, and extend both the Rayleigh quotient and unitary convolution framework to meshes (Sec. 5).

3. Empirically analyze the smoothness behavior of various GNN architectures on complex dynamical systems, showing that controlling smoothness can match or outperform strong baselines (Sec. 6).

## 2. Related Works

**Oversmoothing and undersmoothing in GNNs.** Like Kiani et al. (2024), our work quantifies the effect of neural networks on the Rayleigh quotient (Chung, 1997) of a graph. Kiani et al. (2024) prove that unitary functions, and in particular the unitary convolution network, strictly preserve the Rayleigh quotient and therefore the smoothness of input graphs. However, we show both theoretically and empirically how this property can be overconstraining in GNNs. Other approaches to quantifying smoothness in PDE solutions have used the Matérn kernel (Borovitskiy et al., 2021; Daniels et al., 2025) or decay rate exponents (Kulick et al., 2025), but none have considered the Rayleigh quotient for dynamics models as we do.

Our work is perhaps most similar to Keriven (2022), who also point out that some smoothing can be useful for certain regression tasks but do not consider dynamics modeling specifically. Similarly, Li et al. (2018) point out that

GCNs (Kipf & Welling, 2017) can be understood as a special case of Laplacian smoothing, which is a key reason why GCNs work at all. In fact, Kipf & Welling (2017) argue that their architecture can be understood as a differentiable and parameterized generalization of the 1-dim Weisfeiler-Lehman algorithm (Leman & Weisfeiler, 1968), indicating that even randomly initialized GCNs can be performant due to the way they smooth information throughout the network. Despite these findings, there is comparatively less work studying the role of smoothness in spatio-temporal modeling tasks. While Marisca et al. (2025) study issues with message-passing based GNNs for spatio-temporal modeling, they focus on *oversquashing*, where information fails to propagate to distant nodes, whereas our work addresses *oversmoothing* and *undersmoothing*.

**Dynamics modeling over graphs and meshes.** Our work focuses on dynamics modeling where PDE solutions are discretized as signals on graphs and meshes through the lens of smoothness. Many physical systems, such as wave propagation (d'Alembert, 1747), heat diffusion (baron de Fourier, 1822), phase fields (Cahn & Hilliard, 1958; Li et al., 2024), fluid flows (Constantin & Foiaş, 1988; Anandkumar et al., 2020), and climate systems (Ghil & Simonnet, 2020) can be described by systems of PDEs. Deep learning based approaches are increasingly used to solve these PDEs in these domains where numerical solving is difficult (Wang et al., 2020; Cranmer et al., 2020; Anandkumar et al., 2020; Li et al., 2021; Mustafa et al., 2021; Cai et al., 2021; Maurizi et al., 2022; Park et al., 2023; Liu et al., 2024; Yu & Wang, 2024; Daniels & Rigollet, 2025). For PDE solving on meshes, these dynamics can be formulated extrinsically by embedding the manifold into Euclidean space (Satorras et al., 2021; Pfaff et al., 2021), or intrinsically by defining evolution directly in the coordinates of local tangent spaces (Cohen et al., 2019; de Haan et al., 2021; Mitchel et al., 2021; 2022; Basu et al., 2022; Park et al., 2023; Suk et al., 2024; Mitchel et al., 2024). While Cohen et al. (2019), Pfaff et al. (2021), and Suk et al. (2024) contain isolated experiments related to dynamics modeling, only our work and Park et al. (2023) study how the choice of Euclidean versus locally defined coordinate representations in the network can affect convergence to PDE solutions. Furthermore, our work is distinct from Park et al. (2023) in that only we directly assess how these design choices affect neural network *smoothing behavior*.

**Benchmarking PDE Surrogate Models.** While the physical symmetries of many dynamical systems are well understood (Olver, 1993; Wang et al., 2021; Borovitskiy et al., 2021), smoothness has received less attention. The performance of deep dynamics models is typically measured either via quantitative error metrics against the ground truth or their preservation of underlying physical laws, such as

spectral energy errors (Wang et al., 2021) or equivariance errors (Wang et al., 2021; 2022a;b). Our work is novel in our application of the Rayleigh quotient in quantifying the smoothing effect of trained GNN dynamics models. Furthermore, we are among the first to design architectures with inductive biases that encourage the model to match the Rayleigh quotient of the labeled graphs. Other works have explored using the Rayleigh quotient as an auxiliary loss (Rowan et al., 2025), as positional encodings (Dong et al., 2024), and as a hard constraint to preserve smoothness regardless of the true labels (Kiani et al., 2024). In contrast, only our work and the work of Shao et al. (2024) use architectural inductive biases to match the smoothness of labeled graphs, and only our work evaluates how well the true smoothness of dynamical systems is recovered.

# 3. Background

We first recall the definition of the Rayleigh quotient, a measure of smoothness on graphs, and provide background on unitary convolutions and their invariance to the Rayleigh quotient. We also introduce the mesh data type, which we later use to extend the unitary convolution framework from graphs to meshes.

## 3.1. Rayleigh quotient

To measure the smoothness of a signal on a graph, we use the Rayleigh quotient as defined in Chung (1997).

**Definition 1** (Rayleigh quotient, (Chung, 1997)). Given an undirected graph $\mathcal{G} = (V, E)$ with $|V| = n$ nodes and adjacency matrix $\mathbf{A} \in \{0,1\}^{n \times n}$, let $\mathbf{D} \in \mathbb{R}^{n \times n}$ be a diagonal matrix where the $i$-th entry $\mathbf{D}_{ii} = d_i$ the degree of node $i$. Let $s \colon V \to \mathbb{C}^d$ be a function from nodes to features. Denote by $\tilde{\mathbf{A}} = \mathbf{D}^{-1/2}\mathbf{A}\mathbf{D}^{-1/2}$ the normalized adjacency matrix and $\mathbf{X} \in \mathbb{C}^{n \times d}$ a matrix with the $i$-th row set to feature vector $s(i)$. The Rayleigh quotient is

$$R_{\mathcal{G}}(\mathbf{X}) = \frac{1}{2} \frac{\sum_{(u,v) \in E} \left\| \frac{s(u)}{\sqrt{d_u}} - \frac{s(v)}{\sqrt{d_v}} \right\|^2}{\sum_{w \in V} \|s(w)\|^2} \quad (1)$$

or $\mathrm{Tr}\left(\mathbf{X}^\dagger(\mathbf{I} - \tilde{\mathbf{A}})\mathbf{X}\right) \cdot \|\mathbf{X}\|_F^{-2}$ in matrix form. We will often abbreviate the Laplacian as $\mathbf{L} = (\mathbf{I} - \tilde{\mathbf{A}})$.

Intuitively, the Rayleigh quotient measures the mean difference in node features for adjacent nodes. A graph with identical degree-weighted node features has a Rayleigh quotient of zero.

In this work, we define both oversmoothing and undersmoothing of a model $f$ with respect to the Rayleigh quotient of a target signal [1].

---

[1]This is in contrast to prior works which define oversmoothing

**Definition 2** (Relative Oversmoothing and Undersmoothing). Let $\mathbf{A}$, $\mathbf{X}$, and $\mathcal{G}$ be as in Definition 1. Let $\mathbf{Y} \in \mathbb{R}^{n \times d_{\text{out}}}$ be a node feature matrix for a target signal defined on the same graph $\mathcal{G}$. Let $f \colon \mathbb{R}^{n \times d_{\text{in}}} \times \mathbb{R}^{n \times n} \to \mathbb{R}^{n \times d_{\text{out}}}$ be a function that updates the signal defined on the graph. If the predicted signal is smoother than the ground truth ($R_{\mathcal{G}}(f(\mathbf{X}; \mathbf{A})) < R_{\mathcal{G}}(\mathbf{Y})$), then we say that $f$ is *oversmoothing*. If the predicted signal is less smooth than the ground truth ($R_{\mathcal{G}}(f(\mathbf{X}; \mathbf{A})) > R_{\mathcal{G}}(\mathbf{Y})$), then $f$ is *undersmoothing*.

## 3.2. Unitary Convolution

Kiani et al. (2024) define two different models that preserve the Rayleigh quotient using unitary functions, which satisfy $\mathbf{U}^\dagger\mathbf{U} = \mathbf{I}$. In particular, they define the separable unitary convolution

$$f_{\text{UniConv}}^{\text{sep}}(\mathbf{X}; \mathbf{A}) = \exp(\mathbf{iAt})\mathbf{XU}, \quad \mathbf{U}^\dagger\mathbf{U} = \mathbf{I} \quad (2)$$

and the Lie unitary convolution

$$f_{\text{UniConv}}^{\text{Lie}}(\mathbf{X}; \mathbf{A}) = \exp(\mathbf{AXW}), \quad \mathbf{W} = -\mathbf{W}^\dagger \quad (3)$$

where $\exp(\cdot)$ denotes the matrix exponential. We provide further background material on the matrix exponential and its relationship to unitary matrices in Sec. A.1. The authors show that unitary convolutions are mathematically constrained to preserve the Rayleigh quotient:

**Proposition 1** (Invariance of Rayleigh quotient, Proposition 6 in Kiani et al. (2024)). *Given an undirected graph $\mathcal{G}$ on $n$ nodes with normalized adjacency matrix $\tilde{\mathbf{A}} = \mathbf{D}^{-1/2}\mathbf{A}\mathbf{D}^{-1/2}$, the Rayleigh quotient is invariant under normalized unitary or orthogonal graph convolution, i.e. $R_{\mathcal{G}}(\mathbf{X}) = R_{\mathcal{G}}(f_{\text{UniConv}}(\mathbf{X}))$ where $f_{\text{UniConv}}$ is either seperable or Lie.*

## 3.3. Mesh Data

A (triangular) mesh $\mathcal{M}$ consists of a set $(\mathcal{V}, \mathcal{E}, \mathcal{F})$, where $\mathcal{V}$ is a set of vertices, $\mathcal{E} = \{(i, j)\}$ is a set of ordered vertex indices $i$, $j$ connected by an edge, and $\mathcal{F} = \{(i, j, k)\}$ is the set of ordered vertex indices $i, j, k$ connected by a triangular face. The mesh generalizes graphs by including high order connectivity information via the inclusion of faces. We assume that the mesh is a $2$−dimensional manifold embedded in $\mathbb{R}^3$, i.e. a manifold mesh. The definition of the manifold condition for a mesh is given in Definition 10 (Sec. A.7).

# 4. Theory: Unitary Models are Overconstrained

While unitary models on graphs can be useful because they preserve the Rayleigh quotient, this section illustrates how

---

as exponential decay of the Rayleigh quotient over successive GNN layers towards its minimum, e.g. Fesser et al. (2026).

unitary models can be *overly* constrained. In particular, we derive an approximation error lower bound that clarifies the approximation limits of unitary models. We start by establishing our unitary approximation learning framework.

### 4.1. Preliminaries

Let $Z = \mathbb{C}^n$ be a domain with data probability density $p: Z \to \mathbb{R}$. Let $u: \mathbb{C}^n \to \mathbb{C}^n$ be a unitary function and let $f: \mathbb{C}^n \to \mathbb{C}^n$ be the target function. Denote the regression error by

$$\text{err}_{\text{reg}}(u) = \int_Z p(z)\|u(z) - f(z)\|_2^2 dz.$$

**Group Invariance.** Our main result relies on the theory of approximation error for group invariant functions $h$. We review these concepts in detail in Sec. A.4 and provide informal definitions in the paragraph that follows.

A group invariant function $h$ satisfies $h(z) = h(gz)$ for all $g \in G$, $z \in Z$. Let $Gz = \{gz: g \in G\}$ be the orbit of $z$. A *fundamental domain* $F$ of a group $G$ in $Z$ is a set of orbit representatives. The domain $Z$ can be written as the union of conjugates, $Z = \cup_{g \in G} gF$, where the conjugate is defined $gF = \{gz: z \in F\}$. Denote the integrated density on an orbit by $p(Gz) = \int_{Gz} p(z)dz$. Finally, denote the variance of a function $f$ on an orbit $Gz$ by $\mathbb{V}_{Gz}[f]$. The approximation error lower bound for an invariant function is given by the following proposition.

**Proposition 2** (Theorem 4.8 in Wang et al. (2023)). *For a G-invariant function $h$, the regression error is bounded below by* $\text{err} \geq \int_F p(Gz)\mathbb{V}_{Gz}[f]dz$.

Proposition 2 was initially stated for real-valued functions in Wang et al. (2023), but can be applied to complex valued functions without loss of generality. Furthermore, Wang et al. (2023) provide numerical evidence that Proposition 2 is a tight bound.

### 4.2. Unitary Approximation Error Lower Bound

In this subsection, we state our main theoretical result, which demonstrates that unitary neural networks are over-constrained when the norm of the ground truth function has a high angular dependence. Recall the definition of $\text{SU}(n)$, the group of rotations in $\mathbb{C}^n$:

$$\text{SU}(n) = \{U \in \mathbb{C}^{n \times n}: \det(U) = 1\}.$$

We can now give an approximation error lower bound for unitary models. See Sec. A.5 for the proof.

**Theorem 1.** *Let $F$ be a fundamental domain of $\text{SU}(n)$ in $Z$, e.g. $F = \{te: t \in \mathbb{R}_+\}$ where $e$ is a standard basis vector of $\mathbb{C}^n$. The approximation error of $u$ of $f$ has lower bound*

$$\int_Z p(z)\|u(z) - f(z)\|_2^2 dz \geq \int_F p(\|te\|)\mathbb{V}_{Gz}[\|f\|]dz.$$

The proof of Theorem 1 uses the reverse triangle inequality before applying Proposition 2. Intuitively, the fundamental domain enumerates all concentric spheres $S^{2n-1}$ embedded in $\mathbb{C}^n$. Unitary functions are complex valued rotations and reflections that preserve the norm of data points that live on each sphere. The error lower bound is given by the variance of the norm of $f$ averaged over each concentric sphere where the norm of $u$ is constant. Our result suggests that unitary functions can be particularly overly constraining when the norm of $f$ has a high angular dependence.

## 5. Unitary Convolution Constraint Relaxation

Since Theorem 1 informs us that a unitary convolution network may be overconstraining when the ground truth is not perfectly smoothness preserving, we propose two methods for relaxing unitary convolutions and describe how to extend these architectures from graphs to meshes. We name these methods Taylor truncation and compositional smoothness-breaking. The Taylor truncation method allows precise control of the extent of relaxation, whereas the compositional smoothness-breaking method scales more easily with the number of parameters in the network. The Taylor truncation method is especially useful in cases where the true smoothness is known a priori, in which case theoretical results from the literature (see Sec. A.8) can inform what truncation order $\mathbf{T}_{\max}$ is needed to achieve enough relaxation without grid search or hyperparameter tuning. We will use the first relaxation for a motivating experiment in Sec. 6.1 and the second for a more challenging set of tasks in Sec. 6.2. A comparison of the compositional smoothness-breaking and Taylor truncation methods is shown in Fig. 2. We will use $f_{\text{Layer}}$ to define individual layers and SMALL CAPS TEXT to define architectures constructed from those layers.

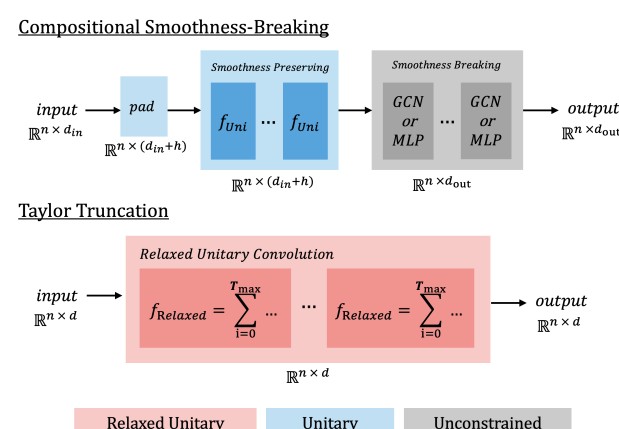

*Figure 2.* **Top:** After zero padding, individual unitary blocks are stacked and the output is fed into an unconstrained smoothness-breaking component. **Bottom:** Each block uses Taylor truncated unitary convolution.

## 5.1. Relaxed Unitary Convolution via Taylor Truncation

We relax Lie unitary convolution by truncating the Taylor series approximations used in Eq. 3. We note that Kiani et al. (2024) do propose their own constraint relaxation for separable unitary convolutions Eq. 2. However, their approach conflates two sources of relaxation: Taylor series truncation of the matrix exponential and letting $\mathbf{U}$ remain unconstrained, making it difficult to measure the individual contributions of each source and to tune the extent of the relaxation. By relaxing the Lie unitary convolution rather than the separable unitary convolution, we isolate the architectural component that alters the Rayleigh quotient. Instead of approximating the matrix exponential using enough Taylor series terms so that the truncation error is negligible, we truncate at some $T = \mathbf{T}_{\max}$ where $\mathbf{T}_{\max}$ controls the extent of the relaxation. Our Taylor-relaxed unitary convolution layer is defined

$$f_{\text{Relaxed}}(\mathbf{X}; \mathbf{A}, \mathbf{T}_{\max}) = \sum_{i=0}^{\mathbf{T}_{\max}} \frac{1}{i!} \mathbf{L}^i(\mathbf{X}) \qquad (4)$$

where $\mathbf{L}(\mathbf{X}) = \mathbf{A}\mathbf{X}\mathbf{W}$, $\mathbf{W} = -\mathbf{W}^\dagger$. While this approach does not preserve the Rayleigh quotient for small $\mathbf{T}_{\max}$, we recover the standard Lie unitary convolution in Eq. 3 in the limit as $T \to \infty$.

Motivated by the desire to find an appropriate $\mathbf{T}_{\max}$ that applies only a small perturbation to the Rayleigh quotients of input graphs, in Sec. B.2 we conduct a sensitivity analysis of the Rayleigh quotient to different Taylor series truncations. Consistent with Kiani et al. (2024), we find that $\mathbf{T}_{\max} = 10$ is sufficient to preserve the Rayleigh quotient. We will refer to models constructed from relaxed layers in Eq. 4 as R-UNIGRAPH.

## 5.2. Relaxed Unitary Convolutional Models via compositional smoothness-breaking

In this section, we note a limitation of relaxed unitary convolution via Taylor truncation that makes it difficult to scale, and propose an alternative relaxation method that addresses this. Since unitary layers cannot change the channel dimension of the node features, the only way to increase the number of parameters in the network is to increase the number of unitary layers. This means that scaling unitary convolutional models requires making the model deeper, which can make training unstable (Balduzzi et al., 2017). As an alternative, we propose a compositional smoothness-breaking method, which subsequently maps the signal through a smoothness preserving layer $P$ and then through a smoothness breaking layer $B$. The smoothness preserving layer first zero-pads the input node features to the desired hidden dimension. This allows us to increase the parameter count without increasing depth. Zero padding also trivially preserves the

Rayleigh quotient since it preserves norms. Concretely, we define our zero padding function $f_{\text{pad}} \colon \mathbb{R}^{n \times d_{\text{in}}} \to \mathbb{R}^{n \times d_{\text{h}}}$ by $\mathbf{X} \mapsto \mathbf{X} \oplus 0$, where $0$ is the $\mathbb{R}^{n \times (d_{\text{h}} - d_{\text{in}})}$ zero matrix. Our approach to increasing the latent dimension is distinct from most prior works on unitary convolution which instead use learnable maps (Kiani et al., 2024; Fesser et al., 2026), though it was hypothesized that zero padding would be an effective approach for approximately unitary models in Kiani et al. (2024). Next, we define our $k$-layer smoothness-preserving component $P$ by

$$P = f_{\text{UniConv}}^{(k)} \circ \cdots \circ f_{\text{UniConv}}^{(1)}(f_{\text{pad}}(\mathbf{X}), \mathbf{A}) \qquad (5)$$

where $f_{\text{UniConv}}$ is either separable or Lie. The smoothness-breaking layers $B$ then serves two purposes: (**i**) map to the target node feature dimension and (**ii**) break the unitary constraint. The smoothness-breaking layer can be any network, such as an MLP or GCN.

## 5.3. Relaxed Unitary Convolution on Meshes

We now generalize unitary convolutional models, relaxed unitary convolutional models, and the mathematical definition of Rayleigh quotient from graphs to meshes. This enables us to solve dynamics problems on manifolds by discretizing them as meshes, such as testing the thermal stability of mechanical parts or weather forecasting on Earth. In particular, we prove that, under modest assumptions on the mesh triangulation, unitary convolution with a weighted adjacency matrix preserves the Rayleigh quotient on meshes (Definition 4); enabling generalization of both unitary and relaxed unitary models to meshes.

**Mesh Rayleigh Quotient.** We first generalize the Rayleigh quotient from graphs to meshes by using the mesh Laplacian instead of the graph Laplacian. The Laplacian on a mesh is typically defined as the symmetric cotangent Laplacian (Reuter et al., 2009) given in Eq. 6, which approximates the Laplace-Beltrami operator for the continuous manifold which the mesh discretizes. For a scalar function $s$ defined on nodes,

$$(\tilde{\mathbf{L}}(s))_i = \frac{1}{2A_i} \sum_{j \in \mathcal{N}(i)} \left( \cot \alpha_{ij} + \cot \beta_{ij} \right) (s_j - s_i) \quad (6)$$

where $\mathcal{N}(i)$ denotes the adjacent vertices of $i$, $\alpha_{ij}$ and $\beta_{ij}$ are the angles opposite edge $(i, j)$, and $A_i$ is the vertex area of $i$. We use the barycentric cell area for $A_i$. We note that it is invalid to define the mesh Rayleigh quotient by replacing $\mathbf{L}$ in Eq. 1 with the symmetric cotangent Laplacian $\tilde{\mathbf{L}}$ in Eq. 6. The cotangent weights in Eq. 6 may be negative, which means that the Rayleigh quotient is no longer a valid measure of smoothness (Definition 1, Rusch et al., 2023). To address this, we use the *Robust Laplacian* (Sharp & Crane, 2020), which performs a minimal edge rewiring of the mesh

so that the cotangent weights obey the *intrinsic Delaunay criterion*.

**Definition 3** (Intrinsic Delaunay Criterion, (Bobenko & Springborn, 2007))**.** For all faces connected by an edge $(i, j)$ with opposite angles $\alpha_{ij}$ and $\beta_{ij}$, $\alpha_{ij} + \beta_{ij} \le \pi$.

Concretely, this means that our Laplacian weights are both symmetric and the off-diagonals are nonnegative; the edge rewiring simply provides an alternative discretization of the same manifold. Denote by $\mathcal{W}$ the cotangent weights

$$\mathcal{W}_{ij} = \begin{cases} \frac{1}{2} \left( \cot \alpha_{ij} + \cot \beta_{ij} \right), & j \in \mathcal{N}(i) \\ - \sum_{k \in \mathcal{N}(i)} \mathcal{W}_{ik}, & i = j \\ 0, & \text{Otherwise.} \end{cases} \tag{7}$$

We define a novel Rayleigh quotient for meshes as follows.

**Definition 4** (Mesh Rayleigh Quotient)**.** Let $\mathcal{M} = (\mathcal{V}, \mathcal{E}, \mathcal{F})$ be a mesh with $|V| = n$ nodes. Denote by $\mathcal{W}$ the cotangent weights corresponding to the Robust Laplacian $\tilde{\mathbf{L}}$. Denote by $\mathcal{E}'$ the rewired edge set given by $\mathcal{E}' = \{(u, v) \colon \mathcal{W}_{uv} \ne 0\}$. Let $s$ and $\mathbf{X}$ be the same as in Definition 1. The mesh Rayleigh quotient is defined

$$R_{\mathcal{M}}(\mathbf{X}) = \frac{1}{2} \frac{\sum\limits_{(u,v) \in \mathcal{E}'} \mathcal{W}_{uv} \left\| \frac{s(u)}{\sqrt{d_u}} - \frac{s(v)}{\sqrt{d_v}} \right\|^2}{\sum\limits_{w \in V} \|s(w)\|^2} = \frac{\mathrm{Tr} \left( \mathbf{X}^\dagger \tilde{\mathbf{L}} \mathbf{X} \right)}{\|\mathbf{X}\|_F^2}.$$

**Unitary Mesh Convolution.** We now make similar modifications to generalize unitary convolution from graphs to meshes. Specifically, we modify the functions in Eq. 2 and Eq. 3 by incorporating the cotangent weights (Eq. 7) into the normalized adjacency matrix $\tilde{\mathbf{A}}$. We note that this edge weighting been shown to improve PDE solving numerically (Crane et al., 2017; Sharp & Crane, 2020) and we are the first to use it in deep dynamics models. In order to prove that incorporating these weights preserves the Rayleigh quotient given by Definition 4, we assume the mesh already satisfies the Delaunay criterion.

**Assumption 1** (Mesh Weights Obey the Delaunay Criterion)**.** For a mesh $\mathcal{M}$, the mesh is manifold and all angles obey the Delaunay Criterion given by Definition 3.

We note that there are existing triangulation strategies that a practitioner can use to ensure that mesh edges satisfy this criterion as a standard data preprocessing step (Huang et al., 2018; Sharp & Crane, 2020), see Sec. A.7 for details. With this assumption, we will now define unitary mesh convolution. Let $\mathbf{D}$ be the weighted degree matrix defined by $\mathbf{D}_{ii} = \sum_{i \ne j} \mathcal{W}_{ij}$. Let $\odot$ represent the Hadamard product which performs element-wise matrix multiplication. Let $\tilde{\mathbf{A}}$ be the normalized adjacency matrix

$\tilde{\mathbf{A}} = \mathbf{D}^{-1/2} \left( \mathcal{W} \odot \mathbf{A} \right) \mathbf{D}^{-1/2}$. We define separable unitary mesh convolution as

$$f_{\text{UniMeshConv}}^{\text{Sep}}(\mathbf{X}; \mathbf{A}, \mathcal{W}) = \exp(i\tilde{\mathbf{A}})\mathbf{X}\mathbf{U} \tag{8}$$

and Lie unitary mesh convolution as

$$f_{\text{UniMeshConv}}^{\text{Lie}}(\mathbf{X}; \mathbf{A}, \mathcal{W}) = \exp(\tilde{\mathbf{A}}\mathbf{X}\mathbf{W}) \tag{9}$$

where $\mathbf{U}\mathbf{U}^\dagger = \mathbf{I}$ and $\mathbf{W} + \mathbf{W}^\dagger = \mathbf{0}$. The following Corollary (proven in Sec. A.6) states that Eq. 8 and Eq. 9 preserve the Rayleigh quotient on meshes.

**Corollary 1** (Corollary to Proposition 1)**.** *Given a mesh $\mathcal{M}$ with normalized adjacency matrix $\tilde{\mathbf{A}} = \mathbf{D}^{-1/2}(\mathcal{W} \odot \mathbf{A})\mathbf{D}^{-1/2}$ that satisfies Assumption 1, the mesh Rayleigh quotient is invariant under normalized unitary or orthogonal graph convolution, i.e. $R_{\mathcal{M}}(\mathbf{X}) = R_{\mathcal{M}}(f_{\text{UniMeshConv}}(\mathbf{X}))$ where $f_{\text{UniMeshConv}}$ is either separable or Lie.*

**Relaxed Unitary Mesh Convolution.** We create a network architecture by coupling the compositional smoothness-breaking relaxation (Sec. 5.2) with Lie unitary mesh convolution in Eq. 9. Concretely, nodes are first zero padded. The smoothness-preserving component $P^{(k)}$ (Eq. 5) is constructed from $k$ layers of Lie unitary mesh convolution (Eq. 9), and a MLP or GCN smoothness-breaking layer $B$ maps to the target. We name our relaxed model R-UNIMESH. R-UNIMESH uses the norm preserving GroupSort activation from Anil et al. (2019) to introduce nonlinearity. R-UNIMESH also uses *orthogonal* weights, since our modeling tasks on meshes in Sec. 6 are real valued.

## 6. Experiments

### 6.1. Motivating Experiment: Heat Flow on Grid Graphs

In this section, we motivate the use of relaxed unitary models by showing that R-UNIGRAPH is able to outperform both normal and Lie unitary graph convolution on predicting heat diffusion. The Taylor truncation method is key to balancing the smoothness preservation of unitary models with the flexibility to capture the true smoothness of the target heat graph.

**Heat Diffusion Dataset.** We use PyGSP (Defferrard et al., 2017) to simulate heat diffusion on $10,000$ two-dimensional grids, each initialized with 20 randomly placed heat sources. Denote by $H \colon \mathbb{R}_+ \to \mathbb{R}^n$ a function that maps time $t$ to the heat distribution of the graph with $n$ nodes. In other words, the heat field on the graph is represented by a feature vector in $\mathbb{R}^{n \times 1}$. In particular, $H(t) = e^{-\tau t \mathbf{L}} H(0)$ where $\tau$ is a diffusivity constant, $\mathbf{L}$ is the graph Laplacian, and $H(0)$ is the initial heat values across the graph. The task is to predict the heat distribution on the graph at time $t = 4$ given

| Model | MSE (↓) | MRE (↓) |
|---|---|---|
| GCN | $1.08 \cdot 10^{-2}$ | $5.99 \cdot 10^{-2}$ |
| Lie Uni | $0.14 \cdot 10^{-2}$ | $8.86 \cdot 10^{-2}$ |
| R-UNIGRAPH (Ours) | $\mathbf{0.11 \cdot 10^{-2}}$ | $\mathbf{2.07 \cdot 10^{-2}}$ |

*Table 1.* MSE and mean Rayleigh quotient error (MRE) of GCN, a Lie unitary convolution network, and R-UNIMESH. The best run for each method out of 5 runs is shown. The best performance is bold.

the heat distribution at time $t = 3$. We denote the target heat field as $\mathbf{Y} = H(4)$. See Sec. B.1 for further dataset details. We compare the performance in terms of MSE loss and mean Rayleigh quotient error for three models: GCN, R-UNIGRAPH, and a Lie unitary model. We use $\mathbf{T}_{\max} = 3$ for the relaxed model. The mean Rayleigh quotient error is given by

$$\mathrm{MRE}(f) = |\overline{R_{\mathcal{G}}(f(\mathbf{X}))} - \overline{R_{\mathcal{G}}(\mathbf{Y})}|.$$

**Results.** As shown in Tab. 1 and Fig. 3, the relaxed model significantly outperforms the GCN and also outperforms the Lie unitary model. Moreover, the relaxed model is best able to produce graphs whose smoothness matches that of the true labels. Our results validate Proposition 7 in Kiani et al. (2024) (also provided in Sec. A.2), which states that GCNs with standard weight initialization tend to increase the smoothness of input signals on the graph as measured by the Rayleigh quotient. Crucially, our experiments reveal that GCNs not only smooth the input signal, but they *oversmooth* compared to the target signal, even when some smoothing is desirable, i.e. the output signal is smoother than the input. In contrast, R-UNIMESH is often initialized in an *undersmooth* state and is able to learn to increase the smoothness of input signals during training to match the smoothness of the target. These insights extend to heat flow on more intricate mesh datasets, as we show in Sec. 6.2.

**Ablation.** We analyze the sensitivity of our method to different $\mathbf{T}_{\max}$ in Sec. B.2, finding that our method is stable across different degrees of relaxation.

### 6.2. Dynamics on Mesh Manifolds

We now consider more challenging and realistic tasks and show that R-UNIMESH is competitive with strong baselines and outperforms all other models on diffusive dynamics. Specifically, our experiments reveal the following practical conclusions: **(i)** R-UNIMESH performs as well as strong baselines such as mesh-aware transformers and equivariant neural networks; **(ii)** R-UNIMESH is especially strong on heat diffusion tasks; **(iii)** geometric inductive biases, such as unitarity or equivariance, are important for strong performance on unseen meshes with complex geometries. We support these conclusions with simulated and real-world

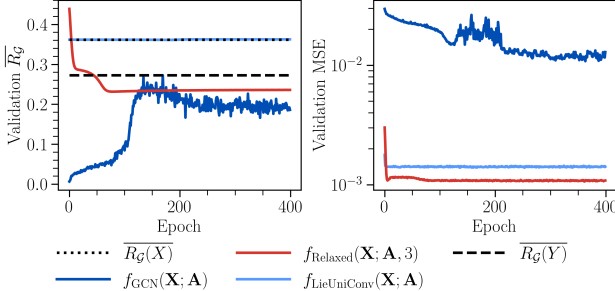

*Figure 3.* **Left:** The dotted lines indicate the mean Rayleigh quotient for input heat graphs at $T = 3$ and target graphs at $T = 4$. We also show the mean Rayleigh quotient for the best performing GCN, R-UNIGRAPH, and Lie unitary models. R-UNIGRAPH is best at capturing the true smoothness. **Right:** Validation MSE of the same three models. R-UNIGRAPH has the best performance. Results for the full set of runs are provided in Sec. B.3.

dynamics datasets on mesh manifolds, including PDE solving on the PyVista (Sullivan & Kaszynski, 2019) meshes from Park et al. (2023) and weather forecasting on the Earth mesh from WeatherBench2 (WB2) (Rasp et al., 2024).

**Baselines.** We include as baselines standard GNN models without any specific inductive biases for working on meshes, including a GCN (Kipf & Welling, 2017) and an MPNN (Gilmer et al., 2017). Additionally, we study symmetry preserving equivariant models, including gauge and Euclidean equivariance (formally defined in Sec. A.3). Informally, Euclidean equivariant models are invariant to roto-translations of the mesh in Cartesian coordinates and Gauge Equivariant GNNs are invariant to a choice of reference angle for models that work in local coordinates of the mesh-manifold. These models have been shown to be particularly strong for PDE solving on meshes (Park et al., 2023). We benchmark a state-of-the-art (SOTA) Euclidean equivariant model (Satorras et al., 2021) as well as different types of Gauge Equivariant GNNs, including convolutional with GemCNN (de Haan et al., 2021), attentional with EMAN (Basu et al., 2022), and message passing with Hermes (Park et al., 2023). We also consider a SOTA mesh transformer (Janny et al., 2023). Finally, we include an operator learning method with MeshGraphNets (Pfaff et al., 2021). We compare these baselines with R-UNIMESH.

**Datasets.** The first task is to auto-regressively predict the solution to the heat, wave, and Cahn-Hilliard equations on test meshes given an initial condition. These equations are defined formally in Sec. C.1. Informally, the heat equation describes the dissipation of temperature. The wave equation describes the propagation of oscillations through a medium. The Cahn-Hilliard equation describes phase separation and gives rise to distinct phase interfaces, for example, the separation of oil and water in a mixture. We use the same

| | Convolutional | | | Attentional | | Message Passing | | |
|---|---|---|---|---|---|---|---|---|
| **Metric** | GCN | GemCNN | R-UniMesh (Ours) | EMAN | Transformer | MPNN | EGNN | Hermes |
| | | | | **Heat** ($\alpha = 1$) | | | | |
| NRMSE ($\downarrow$) | – | – | **51.9 ± 3.6** | 73.50 ± 3.8 | 92.5 ± 5.6 | 99.45 ± 4.8 | 344.25 ± 110.5 | 73.02 ± 4.7 |
| SMAPE ($\downarrow$) | – | 375.4 ± 0.53 | **79.7 ± 5.6** | 110.9 ± 13.3 | 213.9 ± 2.7 | 223.6 ± 1.5 | 319.33 ± 7.59 | 107.6 ± 7.4 |
| RE ($\downarrow$) | – | 52.21 ± 9.4 | **9.1 ± 7.4** | 14.2 ± 1.4 | 46.0 ± 3.7 | 76.06 ± 3.6 | 81.5 ± 8.77 | 39.76 ± 4.7 |
| | | | | **Wave** ($c = 1$) | | | | |
| NRMSE ($\downarrow$) | – | – | **236.5 ± 6.4** | 281.3 ± 15.5 | 864.9 ± 184.9 | 563.6 ± 7.75 | 2280.1 ± 559.9 | 458.5 ± 13.0 |
| SMAPE ($\downarrow$) | – | 318.8 ± 3.9 | 385.2 ± 1.2 | **301.0 ± 4.2** | 327.0 ± 4.4 | 318.0 ± 2.8 | 354.3 ± 11.0 | 316.4 ± 4.5 |
| RE ($\downarrow$) | – | 107.9 ± 3.158 | 93.5 ± 25.4 | 73.57 ± 6.5 | **48.0 ± 7.9** | 139.3 ± 10.1 | 157.2 ± 14.8 | 70.03 ± 6.1 |
| | | | | **Cahn-Hilliard** | | | | |
| NRMSE ($\downarrow$) | – | **121.2 ± 1.8** | 123.9 ± 2.6 | 137.5 ± 0.69 | 144.4 ± 0.8 | 147.4 ± 11.36 | 1001.04 ± 5.73 | 122.0 ± 7.8 |
| SMAPE ($\downarrow$) | – | 204.3 ± 2.4 | 167.3 ± 10.6 | **143.7 ± 2.5** | 191.7 ± 2.0 | 201.22 ± 32.79 | 336.5 ± 2.777 | 173.3 ± 4.3 |
| RE ($\downarrow$) | – | **10.68 ± 3.3** | 18.9 ± 10.4 | 48.57 ± 3.49 | 27.42 ± 2.87 | 23.98 ± 1.4 | 41.8 ± 1.997 | 14.38 ± 11.5 |

*Table 2.* NRMSE, SMAPE, and RE averaged over all rollouts on all test meshes for the heat, wave, and Cahn-Hilliard equations. The best values are in bold and second best are underlined. Errors and standard deviations are reported over all test meshes and initializations. Cells with a dash (–) indicate models that do not converge for a given metric. Errors are scaled by ×196, the number of rollout timesteps. R-UniMesh is competitive across all tasks and excels at solving the heat equation on unseen meshes. Models are grouped together by flavor: convolutional, attention, or message passing (Bronstein et al., 2021).

PyVista meshes generated in Park et al. (2023). These meshes are highly intricate and test the models' ability to handle nonlinear dynamics on complicated geometries. We use five different initializations for each test mesh. Further dataset and training details are provided in Sec. C.1 and Sec. C.2. We will refer to this dataset as MeshPDE in the remainder of the paper.

The second task is weather forecasting using WB2 (Rasp et al., 2024), a widely used benchmark for data-driven global weather forecasting based on historic data. Specifically, we auto-regressively predict future weather conditions on Earth given an initial condition. A formal problem statement is in Sec. D.1. We train and evaluate our models on the ERA5 dataset from WB2, which is the curated version of the ERA5 reanalysis data provided by the European Centre for Medium-Range Weather Forecasts (ECMWF) (Hersbach et al., 2020). We use 1.5 (240 × 120) degree spatial resolution data with a 6 hour temporal resolution, consistent with the evaluation performed in WB2. Further details on mesh construction can be found in Sec. D.3. We evaluate on two variables, temperature at pressure level 850 (T850) and geopotential at pressure level 500 (Z500). We take data from 2013-01-01 to 2019-12-31 UTC as training data. We use a smaller subset of the ERA5 data that is commonly used for other large scale data-based weather models due to compute constraints, but remain consistent to WB2 in evaluating on data from 2020-01-01 to 2020-12-31.

Rayleigh error (RE) is given by:

$$\text{RE}(f) = \frac{1}{\mathbf{T}_{\max}} \sum_{t}^{\mathbf{T}_{\max}} |R_{\mathcal{M}}(\mathbf{Y_t}) - R_{\mathcal{M}}(f(\mathbf{X_t}))|.$$

Further details on these metrics are provided in Sec. C.3 and we compare RE with more multiscale and local smoothness metrics in Sec. C.5 and Sec. C.6. We supplement these metrics with qualitative diagnostic figures in Sec. C.4 and with videos on our GitHub repository, showing in particular that Rayleigh errors are consistent with visual assessments of smoothness. For WB2, we report the root mean squared error (RMSE) and the anomaly correlation coefficient (ACC), both latitude weighted as recommended by the benchmark authors. RMSE measures forecast accuracy, while ACC is the Pearson correlation coefficient between forecast anomalies and ground-truth anomalies relative to a climatological baseline. Precise definitions are provided in Sec. D.

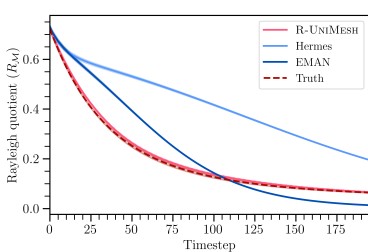

*Figure 4.* The Rayleigh quotient for each timestep on an unseen mesh for Hermes, EMAN, and R-UniMesh models. The R-UniMesh is the best at capturing the true smoothness for heat.

**Evaluation.** For MeshPDE, our metrics include normalized root mean squared error (NRMSE), symmetric mean absolute percentage error (SMAPE), and Rayleigh quotient errors aggregated over all time-steps. In particular, the

**Results.** Our main result is that our R-UniMesh model outperforms all baselines at solving highly diffusive PDEs and captures the true smoothness while remaining competitive across all other tasks. On MeshPDE, as shown in Tab. 2, R-UniMesh performs particularly well on heat modeling,

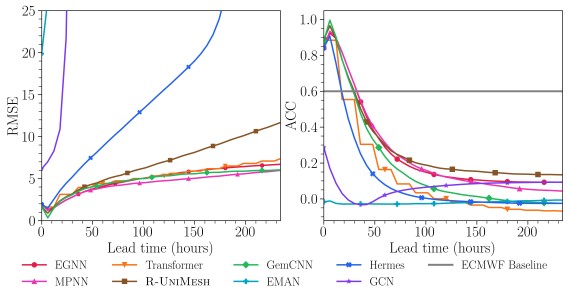

*Figure 5.* RMSE and ACC as a function of lead time for all models temperature prediction. R-UNIMESH has a competitive RMSE, especially at early lead time. R-UNIMESH also maintains viability for lead times of roughly 2 days according to the ECMWF baseline. Exact values recorded in Tab. 13 (Sec. D.4).

where it achieves the lowest NRMSE, SMAPE, and RE. On heat, R-UNIMESH matches the true smoothness almost exactly at every timestep, as seen in Fig. 4, illustrating that R-UNIMESH is best at capturing the underlying differential structure of the PDE solution. The convergence and smoothness errors reported by our metrics also agree visually with our qualitative diagnostics in Fig. 1 and in Sec. C.4.

Another important finding is that nearly all models are able to perform comparably well on the Cahn-Hilliard equation, where the test mesh (toroid) is simple. The only exceptions are GCN and EGNN, which struggle across all tasks. This suggests that stronger geometric inductive biases such as unitarity or gauge equivariance are necessary for strong performance on unseen meshes with complex geometries.

In addition to the attention and message-passing based baselines, R-UNIMESH also outperforms operator learning methods on diffusive dynamics. We provide this comparison in Sec. C.7.

On the real-world WB2 benchmark (Fig. 5), we see that our best performing models are comparable with the state of the art (Rasp et al., 2024) in RMSE and ACC, coming within a couple of degrees of SOTA even at 10 lead days. This is despite restricting our training set size and model parameters due to compute limitations. We show in Sec. D.4 that R-UNIMESH is among the best models for the geopotential prediction task, though all models are below the SOTA in Rasp et al. (2024). Our results also support our earlier finding that geometric inductive biases matter for complex geometries: the equivariant and unitary models show no significant advantages in this setting, where there is no cross mesh generalization. Finally, we show in Sec. D.5 that R-UNIMESH is competitive in terms of RE on both WB2 variables.

**Ablation.** Given that R-UNIMESH is not just relaxed unitary convolution and contains other various architectural considerations such as the choice of readout layers in $B$ and

choice of activation function, we conduct thorough ablation study in Sec. C.8 in order to isolate the effect of the proposed relaxation mechanism and find that it is necessary for faithful modeling. Most importantly, we show that replacing the unitary convolution layers with ordinary graph convolutions causes a significant detriment to performance on diffusive dynamics.

# 7. Discussion

**Limitations.** We note the following limitations in our study. On MeshPDE, our results are strongest for the heat equation, whereas the performance gains are more mixed for wave and Cahn-Hilliard. For wave, we hypothesize that this is because the Rayleigh quotient for wave is highly oscillatory in time. This causes unitarity to be over constraining and our model is therefore more heavily reliant on the unconstrained component to approximate the true dynamics. For Cahn-Hilliard, we hypothesize that our mixed results are because *global* smoothness metrics do not fully capture the underlying dynamics. In particular, the Cahn-Hilliard solution contains *local* and sharp discontinuities in the PDE solution. We note that our experiments on WB2 are reduced in scale and are not intended as a serious claim to SOTA weather forecasting performance. Rather, we use this application as a testbed to study the behavior and inductive biases of our models. On the theoretical side, our bound in Theorem 1 requires knowledge of the data distribution and the ability to write the true function $f$ in angular coordinates in order to be computed on real datasets.

**Conclusion.** Our work clarifies the approximation limits of smoothness preserving (unitary) functions and unitary convolution networks and shows how constraint relaxations can aid performance on various dynamics modeling tasks on graphs and meshes. We contribute R-UNIGRAPH and R-UNIMESH, which provide SOTA performance on diffusive dynamics problems and excel at capturing the true smoothness of the system. Future work will explore using approximately unitary networks for solving PDEs under partial observability by using them as backbones for generative models. Additionally, incorporating boundary conditions into the PDE-solving framework would significantly broaden applicability in engineering settings. Finally, an interesting direction involves the usage of a dynamic $\mathbf{T}_{\max}$ in the Taylor-truncation method, which could yield improvements over a static $\mathbf{T}_{\max}$ in systems with oscillatory smoothness, such as the wave equation.

## Acknowledgements

E.B. Thanks Melanie Weber, Lukas Fesser, Bobak Kiani, and members of the Geometric Machine Learning Group for helpful discussions. L.L. was supported by a Northeastern University Undergraduate Research and Fellowships PEAK Experiences Award. R.W. would like to acknowledge support from NSF Grants 2442658 and 2134178. This work is supported by the National Science Foundation under Cooperative Agreement PHY-2019786 (The NSF AI Institute for Artificial Intelligence and Fundamental Interactions, http://iaifi.org/). We also thank Northeastern Research Computing for GPU access through the Explorer cluster. This paper was reviewed on OpenReview at https://openreview.net/forum?id=Uf3SMkP2Wd.

## Impact Statement

This paper presents work whose goal is to advance the field of Machine Learning. There are many potential societal consequences of our work, none which we feel must be specifically highlighted here.

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

## Appendix Table of Contents

## A. Deferred Theory

This section provides both theoretical background and deferred proofs from the main text.

### A.1. Lie Algebras and the Exponential Map

In this section we review the formalism behind Lie algebras and the exponential map. A group is a mathematical structure that formalizes what it means for something to be *symmetric*. We say that a group is a matrix *Lie group*, if it is a differentiable manifold and a subgroup of the set of invertible $n \times n$ matrices. Lie groups are equipped with a *Lie algebra*, which is the tangent space at the identity element. Our work encounters the orthogonal and unitary Lie groups

$$\mathrm{O}(n) = \{O \in \mathbb{R}^{n \times n} \colon OO^T = I\}, \qquad \mathrm{U}(n) = \{U \in \mathbb{C}^{n \times n} \colon UU^\dagger = I\}$$

as well as the special unitary group

$$\mathrm{SU}(n) = \{U \in \mathbb{C}^{n \times n} \colon \det(U) = 1\}.$$

The associated Lie algebras for $O(n)$ and $U(n)$ are given by

$$\mathfrak{o}(n) = \{M \in \mathbb{R}^{n \times n} \colon M + M^T = 0\}, \qquad \mathfrak{u}(n) = \{M \in \mathbb{C}^{n \times n} \colon M + M^\dagger = 0\}.$$

The exponential map provides a mechanism of parameterizing Lie groups with elements in the Lie algebra. For matrix Lie groups, the exponential map is simply the matrix exponential:

$$\exp(\mathbf{X}) = \sum_i^\infty \frac{1}{i!} \mathbf{X}^i.$$

Applying the exponential map to a linear operator is given by

$$\exp(\mathbf{L})(\mathbf{X}) = \sum_i^\infty \frac{1}{i!} \mathbf{L}^i(\mathbf{X}) = \mathbf{X} + \mathbf{L}(\mathbf{X}) + \frac{1}{2} \mathbf{L} \circ \mathbf{L}(\mathbf{X}) + \frac{1}{6} \mathbf{L} \circ \mathbf{L} \circ \mathbf{L}(\mathbf{X}) + \dots$$

In the case of Eq. 3, $\mathbf{L}$ is graph convolution, $\mathbf{L}(\mathbf{X}) = \mathbf{AXW}$. Further background on group theory and abstract algebra can be found in Artin (1998), Hall (2013), and Esteves (2020).

### A.2. Convolutional oversmoothing

This section provides a result from Kiani et al. (2024) which establishes that Graph Convolution Networks (Kipf & Welling, 2017) have a high probability to exhibit smoothing.

**Proposition 3** (Proposition 7 in Kiani et al. (2024))**.** *Given a simple undirected graph $\mathcal{G}$ on $n$ nodes with normalized adjacency matrix $\widetilde{\mathbf{A}} = \mathbf{D}^{-1/2} \mathbf{A} \mathbf{D}^{-1/2}$ and node degree bounded by $D$, let $\mathbf{X} \in \mathbb{R}^{n \times d}$ have rows drawn i.i.d. from the uniform distribution on the hypersphere in dimension $d$. Let $f_{conv}(\mathbf{X}) = \widetilde{\mathbf{A}} \mathbf{X} \mathbf{W}$ denote convolution with orthogonal feature transformation matrix $\mathbf{W} \in O(d)$. Then, the event below holds with probability $1 - \exp(-\Omega(\sqrt{n}))$:*

$$R_\mathcal{G}(\mathbf{X}) \geq 1 - O\left(\frac{1}{n^{1/4}}\right) \quad and \quad R_\mathcal{G}(f_{conv}(\mathbf{X})) \leq 1 - \frac{\mathrm{Tr}(\widetilde{\mathbf{A}}^3)}{\mathrm{Tr}(\widetilde{\mathbf{A}}^2)} + O\left(\frac{1}{n^{1/4}}\right).$$

## A.3. Gauge and Euclidean Equivariance

In this section, we introduce the necessary background and formal definitions for the equivariance constraints commonly applied to tasks defined on meshes. While working with arbitrary meshes, many commonly used network architectures compute distances between node positions. One has the option of computing these distances in either global Cartesian coordinates or in local tangent spaces of the mesh. In both cases, we may exploit the symmetry of these coordinate systems by enforcing equivariance with respect to transformations from a certain symmetry group into the network architecture, which allows the network to automatically generalize across orbits.

We now give precise definitions of equivariance and invariance.

**Definition 5.** Let $f : \mathcal{X} \to \mathcal{Y}$ be a map between input and output vector spaces $\mathcal{X}$ and $\mathcal{Y}$. Let $G$ be a group with representations $\rho^{\mathcal{X}}$ and $\rho^{\mathcal{Y}}$ which transform vectors in $\mathcal{X}$ and $\mathcal{Y}$ respectively. Representations are group homomorphisms which map group elements to invertible linear transformations. The map $f : \mathcal{X} \to \mathcal{Y}$ is *equivariant* if

$$\rho^{\mathcal{Y}}(g)f(x) = f(\rho^{\mathcal{X}}(g)x) \text{ , for all } g \in G, x \in \mathcal{X} \text{ .}$$

Invariance is a special case of equivariance in which $\rho^{\mathcal{Y}} = \mathrm{Id}^{\mathcal{Y}}$ for all $g \in G$. With an invariant operator, the output of $f$ is unaffected by the transformations applied to the input.

**Definition 6.** A map $f : \mathcal{X} \to \mathcal{Y}$ is *invariant* if

$$f(x) = f(\rho^{\mathcal{X}}(g)x) \text{ , for all } g \in G, x \in \mathcal{X}.$$

### A.3.1. EUCLIDEAN EQUIVARIANCE

For a mesh defined over a global coordinate system, a common choice of symmetry constraint is equivariance to the Euclidean group in $n$ dimensions, $E(n)$. In this setting, the mesh is treated as a graph with positional encodings, and the equivariance constraint ensures generalization to different roto-translations of the mesh.

**Definition 7.** Let $t \in \mathbb{R}^n$ be a translation vector and $Q \in \mathbb{R}^{n \times n}$ an orthogonal matrix representing a rotation or reflection. A function $f$ is equivariant to the Euclidean group $E(n)$ if for any $t \in \mathbb{R}^n$ and $Q \in \mathbb{R}^{n \times n}$ we have

$$f(Qx + t) = Qf(x) + t.$$

### A.3.2. GAUGE EQUIVARIANCE

We may also choose to embed coordinates locally, using coordinates that are intrinsic to the 2D mesh rather than the extrinsic 3D coordinates of the embedding space. This approach arises from the desire for a general convolution-like operator over arbitrary manifolds discretized as a mesh. To encode data over a mesh it is still necessary to make a choice of local coordinate frame at each vertex. In order to guarantee the equivalence of the features resulting from different choices of reference frames, the model should be invariant to change of coordinates frame at each vertex, i.e. gauge equivariant.

We specifically adapt the strategy described in de Haan et al. (2021) and define the local coordinate frame at each vertex in terms of a reference neighboring vertex. Denote $v_a$ as the reference neighbor for gauge $A$, in which the neighbors have angles $\theta_A$, and denote $v_b$ as the reference neighbor for gauge $B$ with angles $\theta_B$. Comparing the two gauges, we see that they are related by a rotation of angle $\phi$, so that $\theta_B = \theta_A - \phi$. This change of gauge is called a gauge transformation of angle $g := \phi$.

**Definition 8** (Equations 3 and 4 in de Haan et al. (2021)). Let $\rho_{\text{in}}$ and $\rho_{\text{out}}$ be input and output types with dimensions $C_{\text{in}}$ and $C_{\text{out}}$. Let $K_{\text{self}} \in \mathbb{R}^{C_{\text{out}} \times C_{\text{in}}}$ and $K_{\text{neigh}} : [0, 2\pi) \to \mathbb{R}^{C_{\text{out}} \times C_{\text{in}}}$ be two kernels. We say the kernels are *gauge equivariant* if for any gauge transformation $g \in [0, 2\pi)$ and for any angle $\theta \in [0, 2\pi)$ we have

$$K_{\text{neigh}}(\theta - g) = \rho_{\text{out}}(-g)K_{\text{neigh}}(\theta)\rho_{\text{in}}(g), \qquad K_{\text{self}} = \rho_{\text{out}}(-g)K_{\text{self}}\,\rho_{\text{in}}(g).$$

Finally, as features at different nodes live in different tangent spaces and thus have different gauges, it is invalid to sum them directly. Let $f_p$ and $f_q$ be node features of a pair of neighboring nodes $p$ and $q$. Before performing gauge equivariant convolution, we must parallel transport each $f_q$ to $T_pM$ along the mesh edge that connects the two vertices for them to be in the same gauge. For more details, we refer the reader to de Haan et al. (2021).

## A.4. Unitary Learning Framework

This section provides rigorous definitions for the mathematical tools used in the main text and additionally clarifies necessary hypotheses.

We start with the fundamental domain. Assume $X$ has dimension $n$. Let $d$ be the dimension of a generic orbit of $G$ in $X$. Let $\nu$ be the $(n-d)$ dimensional Hausdorff measure in $X$.

**Definition 9** (Fundamental Domain, Definition 4.1 in Wang et al. (2023))**.** A closed subset $F$ of $X$ is called a fundamental domain of $G$ in $X$ if $X$ is the union of conjugates of $F$, i.e., $X = \cup_{g \in G} gF$, and the intersection of any two conjugates has measure 0 under $\nu$.

Next, we note that our proof of Theorem 1 satisfies the integrability assumption on the fundamental domain $F$ and orbits $Gz$ established in Wang et al. (2023):

**Assumption 2** (Integrability Hypothesis, Sec. A in Wang et al. (2023))**.** The fundamental domain $F$ and orbit $Gx$ are differentiable manifolds and the union of all pairwise intersections $\cap_{g_1 \neq g_2}(g_1 F \cap g_2 F)$ has measure zero.

We now provide more formal definitions for $\mathbb{E}_{Gx}[f]$ and $\mathbb{V}_{Gx}[f]$ used in Proposition 2. Denote by $q(z) = \frac{p(z)}{p(Gx)}$ the density of the orbit $Gx$ so that $\int_{Gx} q(z)dz = 1$. The mean and variance of a function $f$ on $Gx$ are given by

$$\mathbb{E}_{Gx}[f] = \int_{Gx} q(z)f(z)dz, \qquad \mathbb{V}_{Gx}[f] = \int_{Gx} q(z)\|\mathbb{E}_{Gx}[f] - f(z)\|_2^2 dz.$$

## A.5. Proof of Main Theorem

We now provide proof of our main theoretical result in the main text. We repeat the theorem here for convenience.

**Theorem 1.** Let $F$ be a fundamental domain of $\mathrm{SU}(n)$ in $Z$. In particular, $F = \{te\colon t \in \mathbb{R}_+\}$ where $e$ is a standard basis vector of $\mathbb{C}^n$. The approximation error lower bound can be expressed as

$$\int_Z p(z)\|u(z) - f(z)\|_2^2 dz \geq \int_F p(\|te\|)\mathbb{V}_{Gz}[\|f\|]dz.$$

*Proof of Theorem 1.* By the reverse triangle inequality,

$$\int_Z p(z)\|u(z) - f(z)\|_2^2 dz \geq \int_Z p(z)\left(\|u(z)\| - \|f(z)\|\right)^2 dz.$$

Notice that $\|u(z)\|$ is invariant under the action of $\mathrm{SU}(n)$ on the sphere $S^{2n-1}$ with radius $\|te\|$ and recall that $\mathrm{SU}(n)$ acts transitively on the sphere. Thus, $F = \{te\colon t \in \mathbb{R}_+\}$ is a valid fundamental domain that indexes each orbit $Gz$, the spheres with radii $\|te\|$. Our theorem then follows from Proposition 2. $\qquad\square$

In the following example, we show how this bound may be computed.

**Example 1** (Variance on the Unit Disk)**.** Denote by $D_2$ the unit disk $D_2 = \{(x,y)\colon x^2 + y^2 \leq 1\}$. Let $Z = \mathbb{R}^2$ be a domain with density

$$p = \begin{cases} \frac{1}{\pi}, & (x,y) \in D_2 \\ 0, & \text{Otherwise.} \end{cases}$$

Denote by $f$ a function in polar coordinates given by

$$f\colon (\theta, r) \to \mathbb{R}^2$$
$$\theta \mapsto (\sin\theta + r, \cos\theta + r).$$

On each orbit, we compute

$$\mathbb{E}_{G_z}[f] = (r,r), \qquad \mathbb{V}_{G_z}[f] = 1, \qquad p(r) = \begin{cases} 2r, & r \leq 1 \\ 0, & \text{otherwise.} \end{cases}$$

The approximation error bound of a unitary function $u$ of $f$ is then

$$\int_Z p(z)\|u(z) - f(z)\|_2^2 dz \geq \int_F p(r)\mathbb{V}_{G_z}[f]dr = \int_0^1 2r(1)dr = 1.$$

**A.6. Unitary Convolution on Meshes**

In this section, we prove Corollary 1 stated in Sec. 5.3 and repeated here for convenience.

**Corollary 1** (Corollary to Proposition 1). *Given a mesh $\mathcal{M}$ with normalized adjacency matrix $\tilde{\mathbf{A}} = \mathbf{D}^{-1/2}(\mathcal{W} \odot \mathbf{A})\mathbf{D}^{-1/2}$ that satisfies Assumption 1, the mesh Rayleigh quotient is invariant under normalized unitary or orthogonal graph convolution, i.e. $R_{\mathcal{M}}(\mathbf{X}) = R_{\mathcal{M}}(f_{\mathrm{UniMeshConv}}(\mathbf{X}))$ where $f_{\mathrm{UniMeshConv}}$ is either separable or Lie.*

Our proof follows the same structure as the proof of Proposition 1 in Kiani et al. (2024) with modifications to account for the weighted adjacency matrix. Namely, we invoke Assumption 1 which ensures that $f_{\mathrm{UniMeshConv}}$ is norm preserving and therefore the the strategy in Kiani et al. (2024) still holds.

*Proof.* We first prove invariance for Eq. 8. By the circulant property of the trace,

$$\mathrm{Tr}\left(\left(\exp(i\tilde{\mathbf{A}})\mathbf{X}\mathbf{U}\right)^{\dagger}\left(\mathbf{I} - \tilde{\mathbf{A}}\right)\left(\exp(i\tilde{\mathbf{A}})\mathbf{X}\mathbf{U}\right)\right) = \mathrm{Tr}\left(\mathbf{X}^{\dagger}\exp(-i\tilde{\mathbf{A}})\left(\mathbf{I} - \tilde{\mathbf{A}}\right)\exp(i\tilde{\mathbf{A}})\mathbf{X}\right).$$

Because $\exp(-i\tilde{\mathbf{A}})$, $\exp(i\tilde{\mathbf{A}})$, and $(\mathbf{I} - \tilde{\mathbf{A}})$ share an eigenbasis, they commute, so

$$\mathrm{Tr}\left(\left(\exp(i\tilde{\mathbf{A}})\mathbf{X}\mathbf{U}\right)^{\dagger}\left(\mathbf{I} - \tilde{\mathbf{A}}\right)\left(\exp(i\tilde{\mathbf{A}})\mathbf{X}\mathbf{U}\right)\right) = \mathrm{Tr}\left(\mathbf{X}^{\dagger}\left(\mathbf{I} - \tilde{\mathbf{A}}\right)\mathbf{X}\right).$$

For the denominator, we need to show that $\|\exp(i\tilde{\mathbf{A}})\mathbf{X}\mathbf{U}\|_F^2 = \|\mathbf{X}\|_F^2$. By Assumption 1 we have that $\mathcal{W}$ is symmetric. Because $\mathbf{A}$ is also symmetric, we have that $i\tilde{\mathbf{A}}$ is skew hermitian and therefore $\exp(i\tilde{\mathbf{A}}) \in SU(n)$. Thus, $\|\exp(i\tilde{\mathbf{A}})\mathbf{X}\mathbf{U}\|_F^2 = \|\mathbf{X}\|_F^2$ and finally $R_{\mathcal{M}}(\mathbf{X}) = R_{\mathcal{M}}(f_{\mathrm{UniMeshConv}}(\mathbf{X}))$.

We now show that Eq. 9 also preserves the Rayleigh quotient. First, we need to show that $\|\exp(\mathbf{A}\mathbf{X}\mathbf{W})\|_F^2 = \|\mathbf{X}\|_F^2$. To do this, we note that Eq. 9 can equivalently be viewed us a function that acts on a vector in $\mathbb{C}^{nd}$. By properties of the Kroneckor tensor product,

$$f_{\mathrm{UniMeshConv}}(\mathbf{X}; \mathbf{A}) = \exp(\mathbf{A}\mathbf{X}\mathbf{W}) \iff \mathrm{vec}\left(f_{\mathrm{UniMeshConv}}(\mathbf{X}; \mathbf{A})\right) = \exp\left(\mathbf{A} \otimes \mathbf{W}^{\mathbf{T}}\right)\mathrm{vec}(\mathbf{X}).$$

Since

$$\left(\mathbf{A} \otimes \mathbf{W}^{\mathbf{T}}\right) + \left(\mathbf{A} \otimes \mathbf{W}^{\mathbf{T}}\right)^{\dagger} = \mathbf{A} \otimes \left(\mathbf{W} + \mathbf{W}^{\dagger}\right)^{T} = 0,$$

we have that $\left(\mathbf{A} \otimes \mathbf{W}^{\mathbf{T}}\right)$ is in the lie algebra of the unitary group and therefore preserves the norm of $\mathrm{vec}(\mathbf{X})$. This holds for any symmetric edge weighting $\tilde{\mathbf{A}} = \mathcal{W} \odot \mathbf{A}$, which is guaranteed by Assumption 1. Thus, $\|\exp(\mathbf{A}\mathbf{X}\mathbf{W})\|_F^2 = \|\mathbf{X}\|_F^2$. Next, note that $\exp\left(\tilde{\mathbf{A}} \otimes \mathbf{W}^{\mathbf{T}}\right)$ commutes with $(\tilde{\mathbf{A}} \otimes \mathbf{I})$. Thus,

$$\mathrm{Tr}\left(f_{\mathrm{UniMeshConv}}(\mathbf{X}; \tilde{\mathbf{A}})^{\dagger}(\mathbf{I} - \tilde{\mathbf{A}})f_{\mathrm{UniMeshConv}}(\mathbf{X}; \tilde{\mathbf{A}})\right)$$
$$= \mathrm{vec}(\mathbf{X})^{\dagger}\exp(\tilde{\mathbf{A}} \otimes \mathbf{W}^{\mathbf{T}})^{\dagger}\left[(\mathbf{I} - \tilde{\mathbf{A}}) \otimes \mathbf{I}\right]\exp(\tilde{\mathbf{A}} \otimes \mathbf{W}^{\mathbf{T}})\mathrm{vec}(\mathbf{X})$$
$$= \mathrm{vec}(\mathbf{X})^{\dagger}\left[(\mathbf{I} - \tilde{\mathbf{A}}) \otimes \mathbf{I}\right]\mathrm{vec}(\mathbf{X}).$$

Multipliying the above by $\|\mathbf{X}\|_F^{-2}$ recovers $R_{\mathcal{M}}(\mathbf{X})$. We conclude that $R_{\mathcal{M}}(\mathbf{X}) = R_{\mathcal{M}}(f(\mathbf{X}))$. $\square$

**Remark 2.** Corollary 1 was applied to convolution with the symmetric cotangent weights in Eq. 7, but the proof extends without loss of generality to any set of symmetric weights.

**A.7. Discrete Differential Geometry**

We provide further details and visualizations for concepts in discrete differential geometry, including the mesh manifold condition, the cotangent Laplacian, and Delaunay criterion. We also review results from the literature that suggests that the mesh edge rewiring algorithm used by the Robust Laplacian is a safe choice for the task of PDE solving. For additional reference on these topics, see Meyer et al. (2003) and Crane et al. (2013).

**The Manifold Condition.**  Our work makes use of the assumption that a given mesh is manifold. We recall the following definition of the mesh manifold condition:

**Definition 10** (Manifold Condition, (Sharp & Crane, 2020))**.**  An interior (or boundary) edge $ij$ is manifold if it is contained in exactly two (or one) triangles; an interior (or boundary) vertex $i$ is manifold if the boundary of all triangles incident on $i$ forms a single loop (or path) of edges.

Alternative definitions for the manifold condition often state that a mesh is manifold if interior vertices have local neighborhoods that are homeomorphic to the unit disk and boundary vertices are homeomorphic to the half disk. We refer specifically to Definition 10 when we talk about a mesh being manifold in the paper.

**Cotangent Laplacian Area Normalization.**  Recall that for a scalar function $s$ on mesh vertices we the cotangent Laplacian is defined as

$$(\tilde{\mathbf{L}}(s))_i = \frac{1}{2A_i} \sum_{j \in \mathcal{N}(i)} \left( \cot \alpha_{ij} + \cot \beta_{ij} \right) (s_j - s_i)$$

where $\mathcal{N}(i)$ denotes the adjacent vertices of $i$, $\alpha_{ij}$ and $\beta_{ij}$ are the angles opposite edge $(i, j)$, and $A_i$ is the vertex area of $i$ and that we use the barycentric cell area for $A_i$. In particular, let $\mathcal{A}_{abc}$ be the area of a triangular face with vertices $abc$ and let $\mathcal{F}(i)$ be the set of faces containing vertex $i$. The barycentric cell area is defined

$$A_i = \sum_{abc \in \mathcal{F}(i)} \mathcal{A}_{abc}/3.$$

Normalization by the cell area was used for the dataset construction in Park et al. (2023), but it is not used in the definition of the Robust Laplacian (Sharp & Crane, 2020). In fact, the cell area introduces asymmetry in the edge weights. This is undesirable, as unitary mesh convolution depends on symmetric edge weights in order to preserve the Rayleigh quotient.

**Cotangent Laplacian Edge Weights and Robust Rewiring.**  As noted in Sec. 5.3, an arbitrary mesh may have negative cotangent weights. These cotangent weights have the following geometric meaning. For vertices $ij$ connected by an edge, we say that the edge is *primal*. For manifold meshes, for each primal edge connecting two triangle there exists a *dual edge* that connects the triangle circumcenters. The cotangent weights correspond to the ratio of the primal and dual edge lengths for vertices $ij$ (Crane et al., 2013). The weights are positive when the angles $\alpha_{ij} + \beta_{ij} \leq \pi$. The Robust Laplacian applies two edge rewiring algorithms sequentially, the tufted cover algorithm and the Delaunay edge flip algorithm. The tufted cover algorithm ensures the mesh is manifold so that the Delaunay edge flip algorithm can be applied. The Delaunay edge flip algorithm then ensures that the mesh satisfies the intrinsic Delaunay criterion (Definition 3). A sample edge flip is illustrated in Fig. 6.

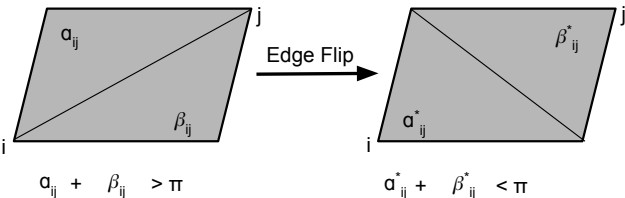

*Figure 6.* Illustration of an intrinsic Delaunay edge flip performed by the Robust Laplacian edge rewiring algorithm. This figure is a reproduced version of Figure 7, Sharp & Crane (2020).

Sharp & Crane (2020) note that from a finite element perspective, changing the triangulation via Delaunay edge-flipping effectively just provides a different set of linear basis functions for the same polyhedral domain. Thus, the only practical concern is whether the tufted cover algorithm, which ensures that the mesh is manifold, does not dramatically change the connectivity. Empirical results from Sharp & Crane (2020) find that the Robust Laplacian greatly *improves* performance computing geodesic distances with the heat method (Crane et al., 2017), which depends on numerical PDE solving on highly nonmanifold meshes with the cotangent Laplacian. This provides confidence that the tufted cover algorithm improves the edge-weighting scheme for unitary mesh convolution. See Crane et al. (2017) for additional reference on numerical PDE solving on nonmanifold meshes.

### A.8. Rayleigh Quotient Sensitivity

We include results from Ferrandi & Hochstenbach (2024) and Dong et al. (2024) that illustrate the sensitivity of the Rayleigh quotient to small perturbations of the input, such as Taylor series truncation errors. While the hypotheses are stronger than what we may actually see in practice, the following proposition provides an intuition for the Rayleigh quotient sensitivity.

**Proposition 4** (Proposition 4 in Ferrandi & Hochstenbach (2024)). *Suppose* $\mathbf{u} = \mathbf{x} + \mathbf{e}$ *is an approximate eigenvector corresponding to a simple eigenvalue* $\lambda \neq 0$ *of a symmetric A, with* $\|\mathbf{x}\| = 1$, $\mathbf{e} \perp \mathbf{x}$, *and* $\varepsilon = \|\mathbf{e}\|$. *Then, up to* $\mathcal{O}(\varepsilon^4)$-*terms, for the sensitivity of the Rayleigh quotient (as a function of* $\mathbf{u}$*) it holds that*

$$\min_{\lambda_i \neq \lambda} \frac{|\lambda_i - \lambda|}{|\lambda_i|} \varepsilon^2 \lesssim \frac{|R_{\mathcal{G}}(\mathbf{u}) - \lambda|}{|\lambda|} \lesssim \max_{\lambda_i \neq \lambda} \frac{|\lambda_i - \lambda|}{|\lambda_i|} \varepsilon^2.$$

This indicates that the Rayleigh quotient sensitivity is quadratic in perturbations $\varepsilon$. For $\varepsilon < 1$, this means that the sensitivity of the Rayleigh quotient is even less than the truncation error. We also have the following results from Dong et al. (2024):

**Proposition 5** (Theorem 1 in Dong et al. (2024)). *For any given graph* $G$, *if there exists a perturbation* $\Delta$ *on* $\mathbf{L}$, *the change of Rayleigh quotient can be bounded by* $\|\Delta\|_2$.

**Proposition 6** (Theorem 2 in Dong et al. (2024)). *For any given graph* $G$, *if there exists a perturbation* $\delta$ *on* $\mathbf{x}$, *the change of Rayleigh quotient can be bounded by* $2\mathbf{x}^T \mathbf{L} \delta + o(\delta)$. *If* $\delta$ *is small enough, in which case* $o(\delta)$ *can be ignored, the change can be further bounded by* $2\mathbf{x}^T \mathbf{L} \delta$.

The results from Dong et al. (2024) state fewer hypotheses than Ferrandi & Hochstenbach (2024). Proposition 5 outlines a bound similar to Proposition 4 in that they are both related to the norm of the perturbing vector. Proposition 6 states an alternative bound related to a perturbation on the input node features instead of the graph's Laplacian. Importantly, one can estimate the truncation error in a unitary convolution layer using Taylor's theorem and substitute this value for $\delta$ in Proposition 6. Comparing the deviation in the Rayleigh quotient with the expected energy dissipation of the PDE gives a model selection criterion for choosing $\mathbf{T}_{\max}$.

# B. Simulated Heat Diffusion Further Details and Results

## B.1. Simulated Heat Diffusion Dataset

This section details dataset generation specifications for our experiment in Sec. 6.1. We generate grid-graphs with an average of 10 nodes and a standard deviation of 2 nodes. On the grid we randomly set 20 nodes to be heat sources. They are given a heat value of 1 and all other nodes start at 0. Using `PyGSP`, We simulate heat flow on 10,000 graphs for training, and the task is to predict the next time step given the previous one. The simulation proceeds until time $T = 10$ in increments of $\Delta T = 0.5$ time steps. A sample graph data point is given in Fig. 7.

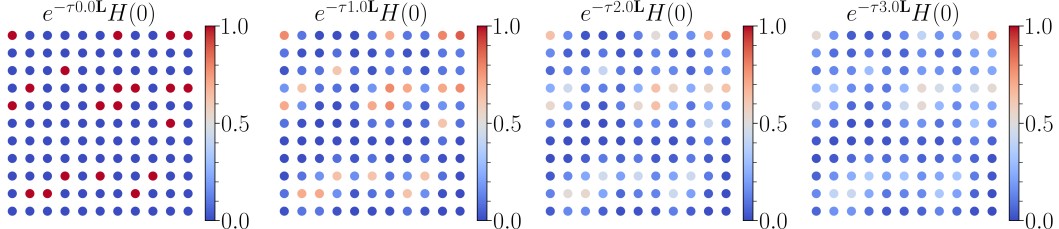

*Figure 7.* Sample heat diffusion process on a grid discretized as a graph. Node neighbors are the nodes that sit adjacent in the grid.

## B.2. Taylor Series Sensitivity Analysis

We conduct a sensitivity analysis of R-UNIGRAPH to different Taylor series truncations. For completeness, we also compare with standard GCNs and Separable unitary networks. First, we study these tendencies at initialization for the heat diffusion dataset that is used for the experiment in Sec. 6.1, described further in Sec. B.1. Our analysis echos a theme similar to Gruver et al. (2023) and Gao et al. (2025) that practitioners should be more thorough in evaluating when numerical approximations break strict theoretical guarantees. Secondly, we study the impact of different truncation terms on downstream performance.

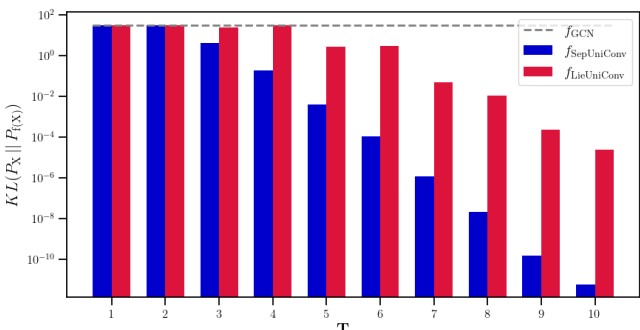

*Figure 8.* KL divergence between distribution of Rayleigh quotients before and after applying the model. Results are averaged over 10 runs.

**Experimental Setup.** We simulate heat diffusion on a grid graph and use time step 3 to conduct the sensitivity analysis. We evaluate on the models $f_{\text{GCN}}$, $f_{\text{SepUniConv}}$, and $f_{\text{LieUniConv}}$. For each model $f$ and truncation length $\mathbf{T}_{\max} \in \{1, \ldots, 10\}$, we compute the Rayleigh quotients $R_{\mathcal{G}}(\mathbf{X})$ and $R_{\mathcal{G}}(f(\mathbf{X}))$ for all graph mini batches $\mathbf{X}$. We denote the distribution of Rayleigh quotients before applying the model by $P_{\mathbf{X}}$ and after applying the model by $P_{f(\mathbf{X})}$. To quantify the deviation between these distributions, we compute the KL divergence $D_{\text{KL}}\left(P_{\mathbf{X}} \parallel P_{f(\mathbf{X})}\right)$, which measures the change in the distribution of Rayleigh quotients caused by the model at initialization. We then examine the impact of these truncations on downstream performance in terms of MSE and Rayleigh error for R-UNIGRAPH.

**Results.** We see in Fig. 8 the effect of Taylor series truncation on the unitarity of the network. In particular, we observe that the KL divergence between the two distributions decreases exponentially with the number of terms. This is to be expected, we know from Taylor's theorem that a truncation at term $t$ gives truncation error $\mathcal{O}\left(\frac{[\|\mathbf{AXW}\|_{\mathbf{O}}]^{t+1}\|\mathbf{X}\|_2}{(t+1)!}\right)$ where $\|\cdot\|_{\mathbf{O}}$ is the

operator norm. Furthermore, works such as Ferrandi & Hochstenbach (2024) and Dong et al. (2024) show theoretically that small truncation errors will not compound into large deviations in the Rayleigh quotient. For details on the relevant propositions from Ferrandi & Hochstenbach (2024) and Dong et al. (2024), see Sec. A.8.

In our GitHub repository we include a video that shows the evolving Rayleigh quotient distribution as we increase $\mathbf{T}_{\max}$. Fig. 8 is also clickable and links to the same video in our anonymous artifact.

We quantify the effect of these truncations on downstream performance in Fig. 9. We see that $\mathbf{T}_{\max} = 3$ achieves the best balance of convergence under both MSE loss and MRE. Moreover, we see that R-UNIGRAPH is not overly sensitive to different choices of $\mathbf{T}_{\max}$ in terms of MSE. Interestingly, the $\mathbf{T}_{\max} = 1$ model still performs well in terms of MSE despite oversmoothing the target signal. The $\mathbf{T}_{\max} = 1$ model can be thought of as a GCN with a hermitian projection of the weights. While the smoothness behavior is similar to what we see for the unconstrained GCN (cf. Fig. 10), the convergence under MSE loss is not. This suggests that there exists a local minimum where the predicted node features are mostly constant that is close to the true heat field. Moreover, it raises the possibility that the benefits of approximately unitary networks lie not only in smoothness preservation but in gradient stability properties. For example, unitary convolutions are known to be perfectly dynamically isometric and 1-Lipschitz (Xiao et al., 2018; Kiani et al., 2024). Networks with these properties tend to avoid vanishing or exploding gradients.

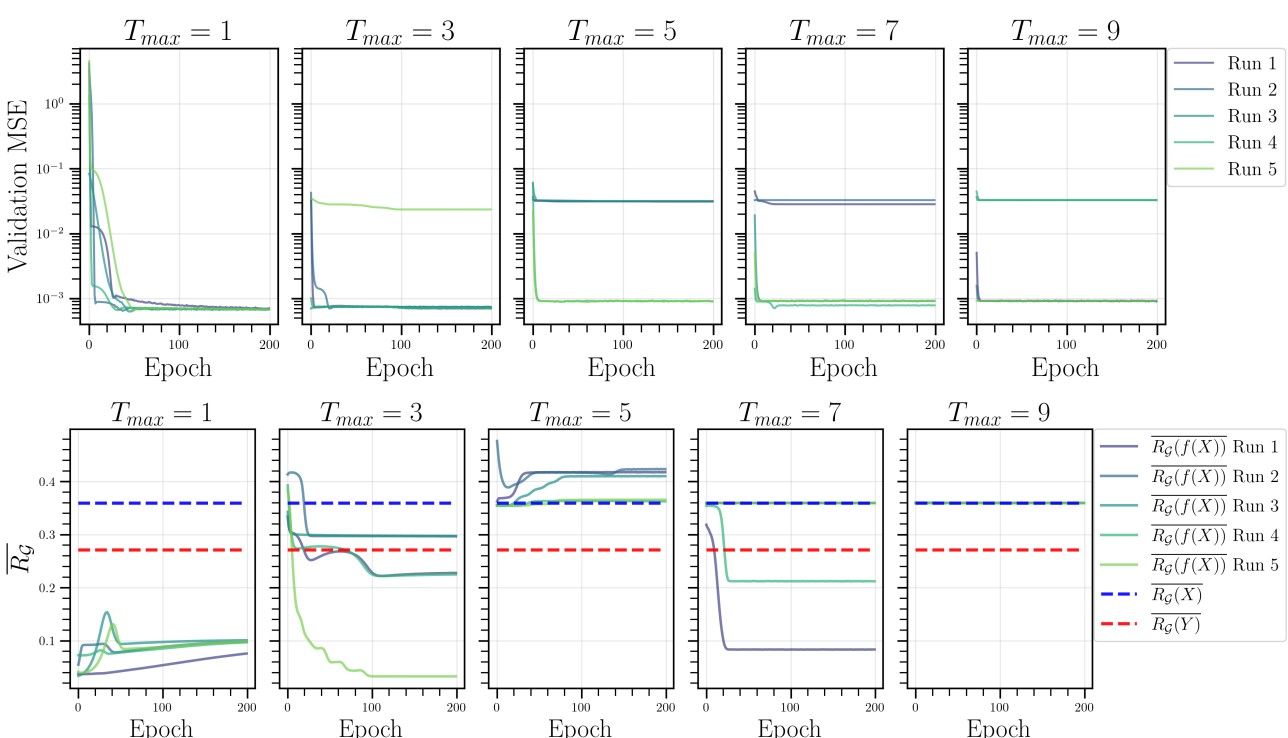

*Figure 9.* Validation MSE and mean Rayleigh quotient for 5 different Lie Unitary Convolution runs. MSE tends to increase as $\mathbf{T}_{\max}$ increases. $\mathbf{T}_{\max} = 3$ offers the best balance between accuracy under MSE loss and smoothness errors.

## B.3. Heat Diffusion Model Ensembles

This section provides the results for Sec. 6.1 for a larger ensemble of models. In Fig. 10, we see that the behavior in the training runs in Fig. 3 occur frequently, with only a couple runs diverging.

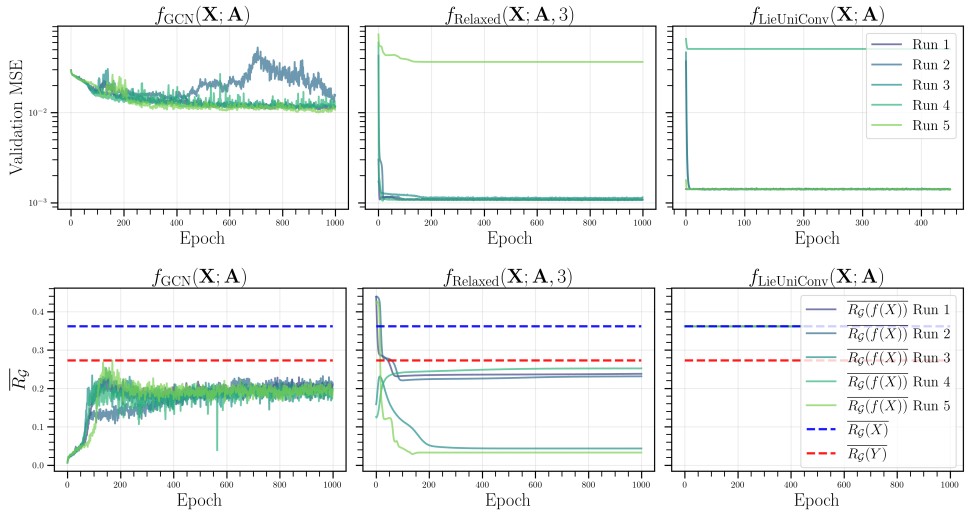

*Figure 10.* **Top:** Validation MSE for an ensemble of 5 runs for a GCN (left), R-UNIGRAPH (middle), and a Lie unitary convolution network (right) at timestep $t = 3$. R-UNIGRAPH significantly outperforms the GCN and also outperforms the Lie unitary network. **Bottom:** The average Rayleigh quotient over all graphs for an ensemble of 5 runs for the same models at timestep $t = 3$. The GCN is under constrained and biased towards oversmoothing at initialization. R-UNIGRAPH is able to roughly match the true smoothness of the labeled graphs. The Lie unitary network is overconstrained and can not model the Rayleigh quotient of the labels because it is forced to preserve the Rayleigh quotient of the input graphs.

# C. MeshPDE Further Details and Results

This section provides extra experimental details and results for our dynamical systems modeling on PyVista meshes in MeshPDE.

## C.1. MeshPDE Dynamical Systems

This section provides details on the PDEs to be solved on the PyVista meshes in MeshPDE. Before defining the PDEs to be solved, let us establish notation. Let $\alpha$ be the thermal diffusivity, and $c$ a constant. The heat and wave equations on the mesh are then given by

$$\frac{\partial u}{\partial t} = \alpha \tilde{\mathbf{L}} u, \quad \text{(Heat)} \qquad\qquad \frac{\partial^2 u}{\partial t^2} = c^2 \tilde{\mathbf{L}} u, \quad \text{(Wave)}.$$

We now define the Cahn-Hilliard equation (Cahn & Hilliard, 1958). Let $c$ be the fluid concentration, $M$ the diffusion coefficient, $\mu$ the chemical potential, $f$ the double-free energy function, and $\lambda$ a positive constant. The Cahn-Hilliard equation is often represented by the following two coupled second order equations:

$$\frac{\partial c}{\partial t} - M \tilde{\mathbf{L}}(\mu) = 0, \quad \mu - \frac{\partial f}{\partial c} + \lambda \tilde{\mathbf{L}}(c) = 0.$$

Here, the double-free energy function is given by $f(c) = 100c^2(1 - c^2)$. Sample initial conditions for each equation on the PyVista meshes are shown in Tab. 11 and Tab. 12 in Sec. C.9.

## C.2. MeshPDE Training Details

We extend the publicly available code base from Park et al. (2023) to train our baselines for the PyVista and WeatherBench2 datasets: https://github.com/jypark0/hermes/. For GemCNN, EMAN, and Hermes, we use the already available pretrained model checkpoints. For GCN, R-UNIMESH, Mesh Transformer, MPNN, and EGNN we train our own models. We use the same train and test splits for the meshes as in Park et al. (2023). Crucially, test meshes are completely held out during training. We performed ablations over learning rate and latent space sizes. Following Park et al. (2023) we keep models within a $\sim 40,000 - 50,000$ parameter budget. We note that this budget is relatively small, and that models that diverge in our experiments could potentially perform better under a more forgiving budget. All runs were performed on a single H200 GPU (NVIDIA, 2025). We use the previous 5 time steps as input node feature vectors and backpropagate through 3 steps of auto-regressive inference.

Hyperparameters are given in our configuration files on GitHub. Defaults are taken from Park et al. (2023) if provided and otherwise optimized via grid search. Considered hyperparameters include learning rate, optimizer, training epochs, latent size, and skip connections. We also consider z-scoring of normed edge lengths for EGNN, different smoothness-breaking heads for R-UNIMESH, and the number of clusters for the mesh transformer.

As observed in Park et al. (2023), we notice that residual connections can be key for performance with the gauge equivariant models. For R-UNIMESH, we found that using a MLP readout with sinusoidal activation functions was a key ingredient for strong performance on MeshPDE. This supports previous work on how to train GNNs for long range tasks (Tönshoff et al., 2023; 2024). However, the GCN head exhibited the best performance on WB2.

## C.3. MeshPDE Evaluation Details

In order to aggregate smoothness errors over all time steps, we introduce a new metric. Define the Rayleigh Error (RE) by $\int_0^\infty |R_{\mathcal{M}}(\mathbf{Y_t}) - R_{\mathcal{M}}(f(\mathbf{X_t}))| dt$. In practice we approximate this by summing over the time steps where we are able to perform inference and normalize to the max timestep:

$$\text{RE}(f) = \frac{1}{\mathbf{T}_{\max}} \sum_t^{\mathbf{T}_{\max}} |R_{\mathcal{M}}(\mathbf{Y_t}) - R_{\mathcal{M}}(f(\mathbf{X_t}))|.$$

Following Janny et al. (2023) and Pandya et al. (2025), we also consider the scale invariant metrics NRMSE and SMAPE averaged over the entire rollout:

$$\text{NRMSE}(f) = \frac{1}{\mathbf{T}_{\max}} \sum_t^{\mathbf{T}_{\max}} \sqrt{\frac{\frac{1}{n} \sum_{i=1}^n \left(f(\mathbf{X_t})_i - (\mathbf{Y_t})_i\right)^2}{\frac{1}{n} \sum_{i=1}^n (\mathbf{Y_t})_i^2}}$$

$$\text{SMAPE}(f) = \frac{1}{\mathbf{T}_{\max}} \sum_t^{\mathbf{T}_{\max}} \frac{1}{n} \sum_{i=1}^n \frac{2|(\mathbf{Y_t})_i - f(\mathbf{X_t})_i|}{|(\mathbf{Y_t})_i| + |f(\mathbf{X_t})_i| + \varepsilon}$$

where $\varepsilon = 10^{-8}$ is a stability constant. SMAPE is generally more robust than NMRSE in that it is less sensitive to outliers, but it is also more sensitive to small values. The scale invariant property of these metrics is crucial especially for heat diffusivity because solutions tend to decrease proportionally to $e^{-t}$. Thus, we need to consider deviations across several orders of magnitude in order to see how accurately we are modeling the decay.

### C.4. MeshPDE Qualitative Diagnostics

In this section, we validate the superior performance of R-UNIMESH on solving the heat equation with qualitative diagnostics. In Tab. 3 we show that R-UNIMESH is the best at capturing the true smoothness of an unseen mesh during each step of the rollout. In our GitHub repository we include a video corresponding to the rollout in Tab. 3 over all timesteps.

### C.5. MeshPDE Multiscale Smoothness Analysis

Since the Rayleigh quotient is a $1-$hop metric, this section performs additional comparisons on MeshPDE with a multiscale smoothness metric and finds that our $1-$hop smoothness tendencies also hold more generally for the gauge equivariant models we study. In particular, we define smoothness according to the $2-$point correlation function. Let $\delta : \mathbb{R}^3 \to \mathbb{R}$ be a function that maps a point $\mathbf{x}$ on a mesh to the scalar solution $u(\mathbf{x})$ (or approximation thereof) to the PDE at that point. The smoothness is then defined by the $2-$point correlation function $\xi$ given in Eq. 10:

$$\xi(r; \delta) = \mathbb{E}\left[\delta(\mathbf{x})\delta(\mathbf{x} + r)\right]. \tag{10}$$

Intuitively, if node features are similar at a distance of $r$ apart, the correlation will be high. This allows us to study smoothness beyond $1-$hop neighbors by considering larger $r$. Fig. 11 shows an example correlation function for Hermes at a given time step. We note that this characterization of smoothness is common in the weak gravitational lensing literature for point-cloud datasets (Schneider, 2006) and are easily computed with the `TreeCorr` library (Jarvis et al., 2004).

Let $\delta_{ij}$ be the scalar field for the ground truth on a mesh $\mathcal{M}_i$ at time step $j$ and $\widehat{\delta_{ij}}$ be the approximation thereof. We define our smoothness error by

$$\text{err}_{\text{smooth}}(\widehat{\delta_{ij}}) = \frac{1}{r_{\text{bins}}} \frac{1}{\mathbf{T}_{\max}} \frac{1}{N} \sum_{i=1}^N \sum_{j=1}^{\mathbf{T}_{\max}} \sum_{k=1}^{r_{\text{bins}}} |\xi(r_k; \delta_{ij}) - \xi(r_k; \widehat{\delta_{ij}})|.$$

We note that the correlation function is related to the Fourier space power spectrum $P(k)$ by

$$\xi(r) = \frac{1}{2\pi^2} \int k^2 P(k) \frac{\sin(kr)}{kr} dk. \tag{11}$$

Thus, Eq. 11 informs us that our metric for smoothness as a function of $r$ is related to traditional energy spectrum errors (e.g., Wang et al., 2021). We leave a more systematic comparison between the measures as an opportunity for future work.

As seen in Tab. 4, the more expressive attention and message passing based models are much better at capturing the underlying smoothness. The CNN model diverges for the heat and wave datasets, but performs reasonably well on Cahn-Hilliard. This is mirrored by our results in Tab. 2 in the main text for the Gauge Equivariant models.

### C.6. MeshPDE Local Smoothness Analysis

This section contains analysis for local smoothness tendencies not captured by global smoothness metrics like the Rayleigh quotient. This is particularly important for the Cahn-Hilliard PDE, where the solution contains sharp discontinuities. To

| Time | Truth | R-UNIMESH **(Ours)** | EMAN | Hermes |
|------|-------|----------------------|------|--------|
| 10 | | | | |
| 50 | | | | |
| 100 | | | | |
| 150 | | | | |
| 190 | | | | |

*Table 3.* Qualitative comparison of model performance for the heat equation on the armadillo mesh. Our R-UNIMESH model remains faithful to the ground truth during each step of the rollout, whereas the EMAN model over smooths and the Hermes model under smooths.

analyze local smoothness, we define smoothness at a given vertex as the Rayleigh quotient of the graph given by the 2-hop neighborhood around that vertex. To quantify how well local smoothness of a surrogate model matches the target, we examine the distribution of Rayleigh quotients for each timestep. The evolving distributions during each timestep of the PDE are available as gifs on our GitHub codebase at `https://github.com/EdwardBerman/rayleigh_analysis/tree/main/rebuttal/localized_metric`. On Cahn-Hilliard, we see that R-UNIMESH is able to capture the local smoothness tendencies on some trajectories, albeit less consistently than GemCNN. This explains the performance in Tab. 2.

## C.7. MeshPDE Operator Learning Method Comparison

We compare R-UNIMESH with MeshGraphNet, a strong operator learning method baseline and show that we outperform on diffusive dynamics. This is shown in Tab. 5.

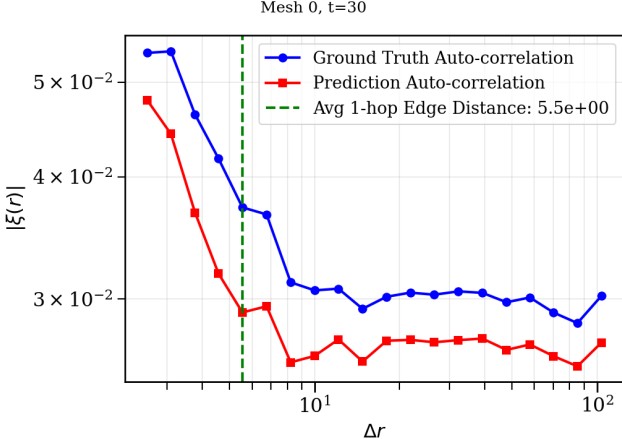

*Figure 11.* Smoothness for a Hermes model as measured by the 2−point correlation function. The plot indicates undersmoothing in each radial bin.

| Heat ($\alpha = 1$) | |
| --- | --- |
| **Model** | $\text{err}_{\text{smooth}}$ ($\downarrow$) |
| GemCNN | – |
| EMAN | $\mathbf{4.04 \times 10^{-3}}$ |
| Hermes | $9.71 \times 10^{-3}$ |
| **Wave ($c = 1$)** | |
| **Model** | $\text{err}_{\text{smooth}}$ ($\downarrow$) |
| GemCNN | – |
| EMAN | $\mathbf{1.78 \times 10^{-3}}$ |
| Hermes | $1.38 \times 10^{-2}$ |
| **Cahn–Hilliard** | |
| **Model** | $\text{err}_{\text{smooth}}$ ($\downarrow$) |
| GemCNN | $1.89 \times 10^{-1}$ |
| EMAN | $4.59 \times 10^{-1}$ |
| Hermes | $\mathbf{9.61 \times 10^{-3}}$ |

*Table 4.* $\text{err}_{\text{smooth}}$ for Gauge Equivariant models on the MeshPDE datasets. Dashes (–) indicate non-convergence. Best performing model is indicated with **bold text**.

| Metric | R-UniMesh | MeshGraphNet |
| --- | --- | --- |
| **Heat ($\alpha = 1$)** | | |
| NRMSE ($\downarrow$) | $\mathbf{51.9 \pm 3.6}$ | $108.1 \pm 3.2$ |
| SMAPE ($\downarrow$) | $\mathbf{79.7 \pm 5.6}$ | $144.1 \pm 6.3$ |
| RE ($\downarrow$) | $\mathbf{9.1 \pm 7.4}$ | $53.4 \pm 1.6$ |
| **Wave ($c = 1$)** | | |
| NRMSE ($\downarrow$) | $\mathbf{236.5 \pm 6.4}$ | $253.6 \pm 28.2$ |
| SMAPE ($\downarrow$) | $385.2 \pm 1.2$ | $\mathbf{278.6 \pm 2.3}$ |
| RE ($\downarrow$) | $93.5 \pm 25.4$ | $\mathbf{30.0 \pm 23.2}$ |
| **Cahn-Hilliard** | | |
| NRMSE ($\downarrow$) | $123.9 \pm 2.6$ | $\mathbf{122.0 \pm 1.5}$ |
| SMAPE ($\downarrow$) | $\mathbf{167.3 \pm 10.6}$ | $186.1 \pm 2.9$ |
| RE ($\downarrow$) | $18.9 \pm 10.4$ | $\mathbf{6.2 \pm 0.4}$ |

*Table 5.* NRMSE, SMAPE, and RE averaged over all rollouts on all test meshes for the heat, wave, and Cahn-Hilliard equations. The best values are in bold. R-UniMesh outperforms MeshGraphNet on heat and is competitive on the other PDEs.

## C.8. MeshPDE Ablation Study

We provide ablations on each of the main components in R-UniMesh and show that are proposed relaxation mechanism is necessary for faithful modeling. Most importantly, Tab. 6 shows that unitarity graph convolutions are necessary for strong performance on diffusive dynamics. While our method can tolerate early Taylor truncations in the unitary layers of

the preserver network $P$, replacing the unitary layers with fully unconstrained GCN layers results in significantly worse performance. Similarly, in Tab. 7 we show that the smoothness preserving zero pad operation is necessary for strong performance across the heat task. However, we see in Tab. 8 that our method is not overly sensitive to our usage of the GroupSort activation function. Finally, we note that our method was not overly sensitive to the width and depth of the decoding MLP used in our experiments. As seen in Tab. 9 and Tab. 10, while our results were best for an MLP with 3 layers and a hidden width of size 64, performance was still competitive with many baselines for a single readout layer or with a width of 128.

| Metric | Unitary GCN ($T = 10$) | Unitary GCN ($T = 3$) | Unconstrained GCN |
|---|---|---|---|
| **Heat** | | | |
| NRMSE ($\downarrow$) | $\mathbf{51.9 \pm 3.6}$ | $70.1 \pm 6.3$ | $416.9 \pm 14.3$ |
| SMAPE ($\downarrow$) | $79.7 \pm 5.6$ | $\mathbf{76.7 \pm 10.4}$ | $248.8 \pm 6.6$ |
| RE ($\downarrow$) | $9.1 \pm 7.4$ | $\mathbf{9.0 \pm 0.6}$ | $31.5 \pm 8.6$ |
| **Wave** | | | |
| NRMSE ($\downarrow$) | $236.5 \pm 6.4$ | $242.2 \pm 4.1$ | $\mathbf{196.0 \pm 0.0}$ |
| SMAPE ($\downarrow$) | $385.2 \pm 1.2$ | $\mathbf{299.1 \pm 5.9}$ | $363.1 \pm 3.1$ |
| RE ($\downarrow$) | $93.5 \pm 25.4$ | $\mathbf{37.7 \pm 4.8}$ | $193.6 \pm 4.3$ |
| **Cahn-Hilliard** | | | |
| NRMSE ($\downarrow$) | $\mathbf{123.9 \pm 2.6}$ | $130.1 \pm 2.2$ | $175.0 \pm 0.9$ |
| SMAPE ($\downarrow$) | $167.3 \pm 10.6$ | $\mathbf{155.9 \pm 7.2}$ | $187.0 \pm 2.8$ |
| RE ($\downarrow$) | $18.9 \pm 10.4$ | $24.2 \pm 8.5$ | $\mathbf{12.0 \pm 4.6}$ |

*Table 6.* Comparison for the smoothness-preserving component type on NRMSE, SMAPE, and RE averaged over all rollouts on all test meshes for the heat, wave, and Cahn-Hilliard equations. The best values are in bold.

| Metric | Zero Pad | GCN |
|---|---|---|
| **Heat** | | |
| NRMSE ($\downarrow$) | $\mathbf{51.9 \pm 3.6}$ | $4{,}689.6 \pm 232.0$ |
| SMAPE ($\downarrow$) | $\mathbf{79.7 \pm 5.6}$ | $367.7 \pm 0.6$ |
| RE ($\downarrow$) | $\mathbf{9.1 \pm 7.4}$ | $153.9 \pm 13.4$ |
| **Wave** | | |
| NRMSE ($\downarrow$) | $\mathbf{236.5 \pm 6.4}$ | $12{,}532.0 \pm 242.2$ |
| SMAPE ($\downarrow$) | $385.2 \pm 1.2$ | $\mathbf{383.9 \pm 0.6}$ |
| RE ($\downarrow$) | $93.5 \pm 25.4$ | $\mathbf{11.8 \pm 2.8}$ |
| **Cahn-Hilliard** | | |
| NRMSE ($\downarrow$) | $\mathbf{123.9 \pm 2.6}$ | $228.5 \pm 1.0$ |
| SMAPE ($\downarrow$) | $\mathbf{167.3 \pm 10.6}$ | $250.8 \pm 1.3$ |
| RE ($\downarrow$) | $18.9 \pm 10.4$ | $\mathbf{9.0 \pm 4.3}$ |

*Table 7.* Comparison of Zero Pad and GCN mapping strategies for increasing the latent dimension on NRMSE, SMAPE, and RE averaged over all rollouts on all test meshes for the heat, wave, and Cahn-Hilliard equations. The best values are in bold.

| Metric | GroupSort | ReLU |
|---|---|---|
| **Heat** | | |
| NRMSE ($\downarrow$) | $\mathbf{51.9 \pm 3.6}$ | $87.7 \pm 10.5$ |
| SMAPE ($\downarrow$) | $\mathbf{79.7 \pm 5.6}$ | $82.9 \pm 3.6$ |
| RE ($\downarrow$) | $9.1 \pm 7.4$ | $\mathbf{9.0 \pm 10.5}$ |
| **Wave** | | |
| NRMSE ($\downarrow$) | $\mathbf{236.5 \pm 6.4}$ | $825.7 \pm 12.3$ |
| SMAPE ($\downarrow$) | $385.2 \pm 1.2$ | $\mathbf{340.2 \pm 4.3}$ |
| RE ($\downarrow$) | $93.5 \pm 25.4$ | $\mathbf{22.4 \pm 4.0}$ |
| **Cahn-Hilliard** | | |
| NRMSE ($\downarrow$) | $\mathbf{123.9 \pm 2.6}$ | $144.7 \pm 4.2$ |
| SMAPE ($\downarrow$) | $167.3 \pm 10.6$ | $151.2 \pm 4.5$ |
| RE ($\downarrow$) | $\mathbf{18.9 \pm 10.4}$ | $30.0 \pm 8.0$ |

*Table 8.* Comparison of GroupSort and ReLU activation functions on NRMSE, SMAPE, and RE averaged over all rollouts on all test meshes for the heat, wave, and Cahn-Hilliard equations. The best values are in bold.

| Metric | MLP (Depth = 1) | MLP (Depth = 3) |
|---|---|---|
| **Heat** | | |
| NRMSE (↓) | 99.9 ± 13.1 | **51.9 ± 3.6** |
| SMAPE (↓) | 181.6 ± 4.0 | **79.7 ± 5.6** |
| RE (↓) | 21.2 ± 7.1 | **9.1 ± 7.4** |
| **Wave** | | |
| NRMSE (↓) | − | 236.5 ± 6.4 |
| SMAPE (↓) | **377.6 ± 4.1** | 385.2 ± 1.2 |
| RE (↓) | 109.6 ± 24.8 | **93.5 ± 25.4** |
| **Cahn-Hilliard** | | |
| NRMSE (↓) | 141.9 ± 1.9 | **123.9 ± 2.6** |
| SMAPE (↓) | 233.7 ± 7.8 | **167.3 ± 10.6** |
| RE (↓) | 25.5 ± 5.2 | **18.9 ± 10.4** |

*Table 9.* Effect of MLP depth on NRMSE, SMAPE, and RE averaged over all rollouts on all test meshes for the heat, wave, and Cahn-Hilliard equations. The best values are in bold.

| Metric | MLP (Width = 64) | MLP (Width = 128) |
|---|---|---|
| **Heat** | | |
| NRMSE (↓) | **51.9 ± 3.6** | 88.2 ± 6.8 |
| SMAPE (↓) | **79.7 ± 5.6** | 136.3 ± 4.7 |
| RE (↓) | **9.1 ± 7.4** | 10.7 ± 2.3 |
| **Wave** | | |
| NRMSE (↓) | **236.5 ± 6.4** | 1,433.0 ± 35.0 |
| SMAPE (↓) | 385.2 ± 1.2 | **343.4 ± 7.7** |
| RE (↓) | **93.5 ± 25.4** | 176.0 ± 4.2 |
| **Cahn-Hilliard** | | |
| NRMSE (↓) | **123.9 ± 2.6** | 170.6 ± 1.0 |
| SMAPE (↓) | 167.3 ± 10.6 | **155.4 ± 1.2** |
| RE (↓) | **18.9 ± 10.4** | 47.3 ± 3.0 |

*Table 10.* Effect of MLP width on NRMSE, SMAPE, and RE averaged over all rollouts on all test meshes for the heat, wave, and Cahn-Hilliard equations. The best values are in bold.

## C.9. MeshPDE Sample Initializations

| Mesh | Heat $T = 0$ | Wave $T = 0$ |
| --- | --- | --- |

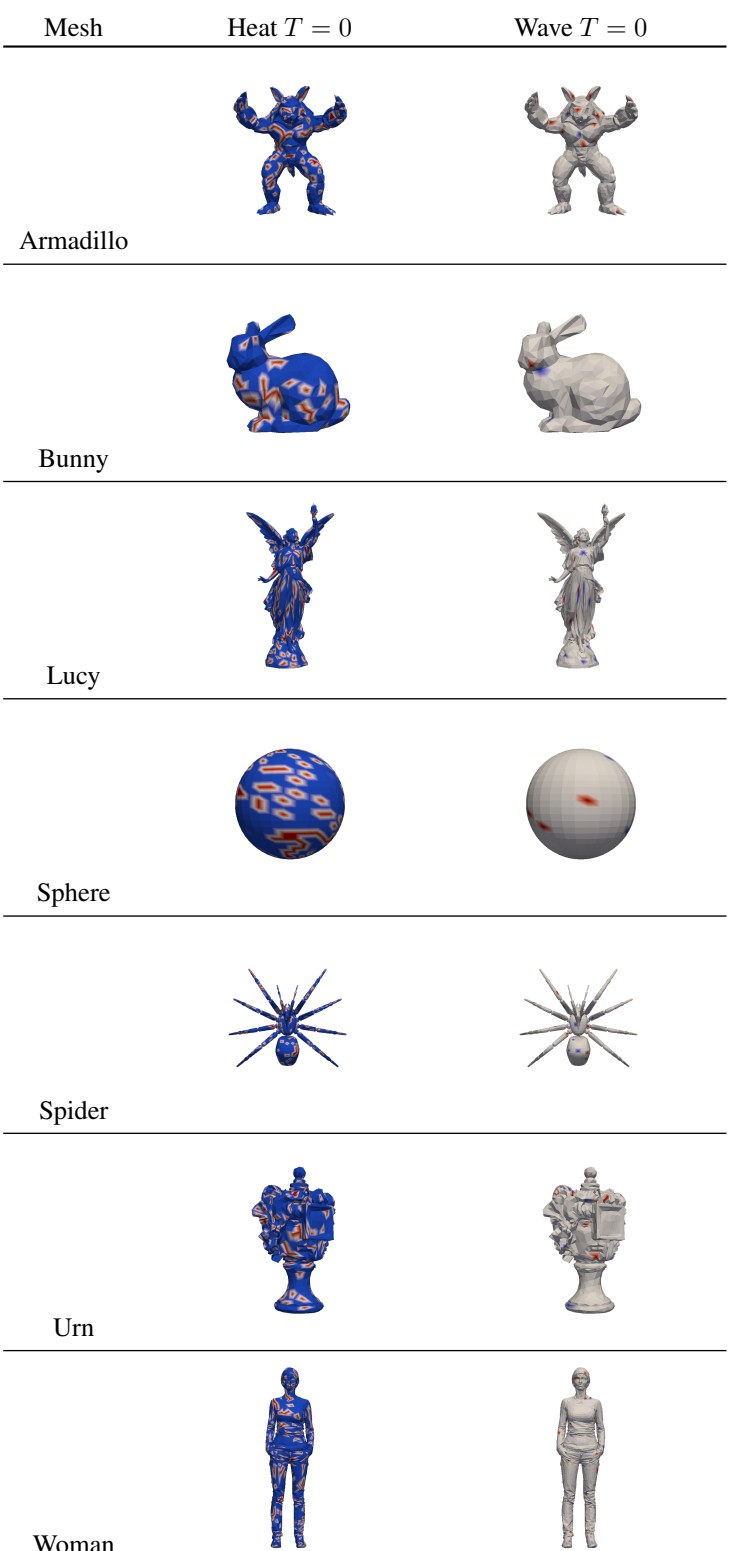

*Table 11.* Sample initializations for the heat and wave equations on the `PyVista` meshes.

| Mesh | Cahn-Hilliard $T = 0$ |
|------|------------------------|

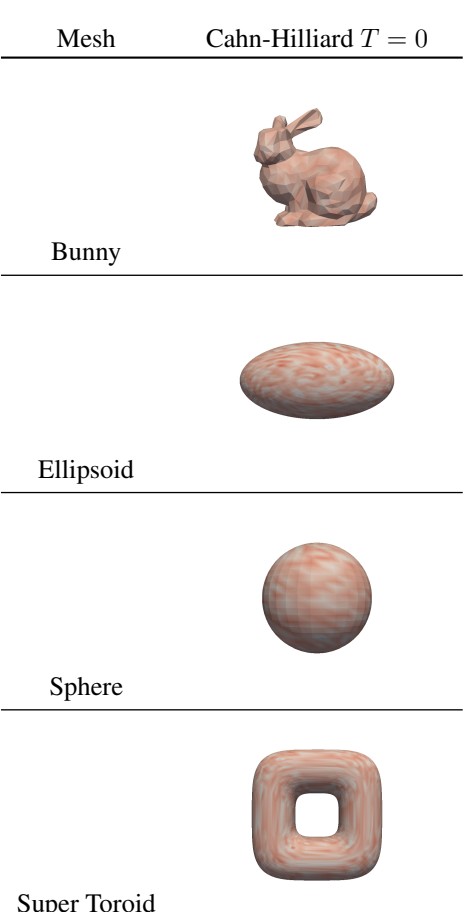

| Bunny | |
|-------|--|

| Ellipsoid | |
|-----------|--|

| Sphere | |
|--------|--|

| Super Toroid | |
|--------------|--|

*Table 12.* Sample initializations for the Cahn-Hilliard equation on the `PyVista` meshes.

# D. WeatherBench2 Further Details and Results

We lay out the relevant training, evaluation, and dataset details for WB2.

## D.1. WB2 Training Details and Problem Statement

Our WB2 problem statement is as follows. Given a dataset $\mathcal{D} = \{X_i\}_{i=1}^N$ of historical weather data, the task of weather forecasting is to predict future weather conditions $X_T \in \mathbb{R}^{V \times H \times W}$ given initial conditions $\{X_i\}_{i=1}^K, X_i \in \mathbb{R}^{V \times H \times W}$, where $T$ is the target lead time, $K$ is the number of input time steps to the model, $V$ is the number of atmospheric variables, and $H \times W$ is the spatial resolution of the data, which depends on how densely we grid the globe. We follow the same training and hyper parameter optimization strategy as in Sec. C.2. The only difference is that we use the 3 previous time steps as input instead of 5. All models are given a consistent compute budget of 8 hours on an NVIDIA H200 GPU for up to 100 epochs.

## D.2. WB2 Evaluation Details

Here we give precise definitions of the evaluation and metrics omitted in the main text. We begin by establishing some notation common to the subsections, and consistent with the notation used in (Rasp et al., 2024).

Let $f$ denote the forecast, $o$ the ground-truth observation, and $c$ the climatology. Let $t \in \{1, \ldots, T\}$ denote the verification time, $l \in \{1, \ldots, L\}$ the lead time, $i \in \{1, \ldots, I\}$ the latitude index, and $j \in \{1, \ldots, J\}$ the longitude index. Forecasts are indexed as $f_{t,l,i,j}$, while observations and climatology are indexed by absolute time as $o_{t,i,j}$ and $c_{t,i,j}$.

### D.2.1. LATITUDE WEIGHTING

In an equiangular latitude-longitude grid, grid cells at the poles have a much smaller area compared to grid cells at the equator. Weighting all cells equally in the computation of RMSE and ACC would result in an inordinate bias towards the polar regions. As a result both metrics are latitude-weighted with weights computed as follows:

$$w(i) = \frac{\sin \theta_i^u - \sin \theta_i^l}{\frac{1}{I} \sum_i^I (\sin \theta_t^u - \sin \theta_i^l)},$$

where $\theta_i^u$ and $\theta_i^l$ indicate upper and lower latitude bounds, respectively.

### D.2.2. CLIMATOLOGY

The climatology $c$ is a function of the day of year and time of day, it is computed by taking the mean of ERA5 data from 1990 to 2019 (inclusive) for each grid point. A sliding window of 61 days is used around each day of year and time of day combination with weights linearly decaying to zero from the center. For notational consistency, we also define the lead-time–indexed climatology $c_{t,l,i,j} := c_{t+l,i,j}$, corresponding to the climatology at the forecast valid time.

### D.2.3. ROOT MEAN SQUARED ERROR (RMSE)

Following the WB2 convention, our work measures error in terms of RMSE. For each variable and level pair, the RMSE at lead time $l$ is defined as:

$$\text{RMSE}_l = \sqrt{\frac{1}{TIJ} \sum_t^T \sum_i^I \sum_j^J w(i)(f_{t,l,i,j} - o_{t,i,j})^2}.$$

This choice is important for temperature forecasting, as we are invariant to choice of unit (e.g., temperature in terms of Kelvin and Celsius will have the same RMSE). Moreover, the change in scale over time is less dramatic as it was for the `PyVista` meshes, where we considered NRMSE.

### D.2.4. ANOMALY CORRELATION COEFFICIENT (ACC)

The ACC is computed as the Pearson correlation coefficient of the anomalies with respect to the climatology $c$. Denote the differences between forecast and climatology and between observation and climatology by

$$f'_{t,l,i,j} = f_{t,l,i,j} - c_{t,l,i,j}; \quad o'_{t,i,j} = o_{t,i,j} - c_{t,i,j}.$$

The ACC at lead time $l$ is then defined as

$$\text{ACC}_l = \frac{1}{T} \sum_t^T \frac{\sum_i^I \sum_j^J w(i) f'_{t,l,i,j} \, o'_{t,i,j}}{\sqrt{\sum_i^I \sum_j^J w(i) f'_{t,l,i,j}}^2 \sum_i^I \sum_j^J w(i) o'_{t,i,j}}^2}.$$

ACC ranges from 1, indicating perfect correlation, to $-1$, indicating perfect anti-correlation. The ECMWF states that when the ACC value falls below 0.6, it is considered that the positioning of synoptic scale features ceases to have value for forecasting purposes.

### D.3. WB2 Earth Mesh Discretization

We construct a spherical mesh of the Earth by directly projecting the latitude–longitude grid points onto the unit sphere, and define mesh connectivity according to the original grid neighborhood structure. In order to obtain triangular faces, we further subdivide each cell into two triangles. The resulting mesh has 29040 nodes, 57600 faces, and 86640 edges. We note that this mesh construction is simpler than those used in other graph-based models, such as GraphCast (Lam et al., 2023), which employs a subdivided icosahedron as the underlying mesh. However, our approach has the advantage that it operates directly on the native latitude–longitude grid and therefore does not require interpolation or regridding of the ERA5 data. Although more elaborate mesh constructions is likely to improve performance in real weather forecasting applications, our focus is on methodological experimentation rather than optimized weather prediction, and we therefore leave mesh optimization as an opportunity for future work.

### D.4. WB2 Temperature and Geopotential Extended Results

We report the RMSE and ACC curves for all lead times. We see in Fig. 12 that R-UNIMESH stays valid for the longest lead time in terms of ACC, and is rivaled only by GemCNN in terms of RMSE for Geopotential. The results of Fig. 5 and Fig. 12 are also given as a table in Tab. 13.

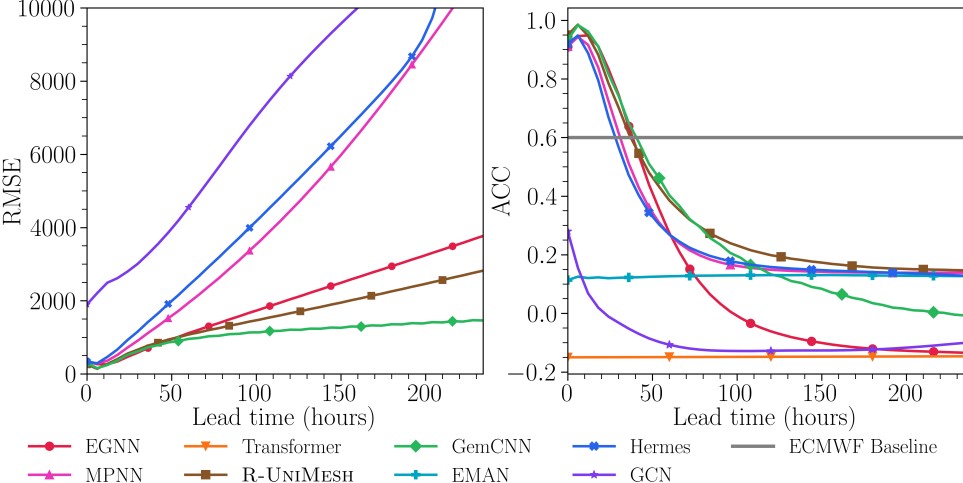

*Figure 12.* RMSE and ACC as a function of lead time for all models geopotential prediction. R-UNIMESH has a competitive RMSE, especially at early lead time. R-UNIMESH also maintains viability for lead times of roughly 2 days according to the ECMWF baseline.

### D.5. WB2 Smoothness Extended Results

We provide smoothness errors for all models on WB2 temperature and geopotential datasets. As seen in Tab. 14, R-UNIMESH is competitive across both temperature and geopotential, and is within statistical significance of the best performing EGNN model on geopotential.

| Model | Metric | 6h | 12h | 18h | 24h | 30h | 36h | 42h | 48h | 240h |
|---|---|---|---|---|---|---|---|---|---|---|
| EGNN | Z500 ACC | 0.909 | 0.946 | 0.948 | 0.912 | 0.835 | 0.747 | 0.639 | 0.538 | -0.134 |
| | Z500 RMSE | 347.83 | 266.96 | 261.09 | 340.43 | 467.68 | 581.20 | 706.01 | 812.70 | 3769.31 |
| | T850 ACC | 0.840 | 0.923 | 0.894 | 0.817 | 0.725 | 0.637 | 0.541 | 0.456 | 0.093 |
| | T850 RMSE | 1.94 | 1.32 | 1.54 | 2.02 | 2.46 | 2.81 | 3.17 | 3.47 | 6.71 |
| EMAN | Z500 ACC | 0.114 | 0.125 | 0.121 | 0.123 | 0.120 | 0.122 | 0.123 | 0.123 | 0.127 |
| | Z500 RMSE | 39530.01 | 52028.53 | 65037.68 | 89210.70 | 103179.47 | 119529.81 | 139314.85 | 156786.23 | 4898180.46 |
| | T850 ACC | -0.016 | -0.011 | -0.025 | -0.029 | -0.029 | -0.028 | -0.028 | -0.028 | -0.007 |
| | T850 RMSE | 19.87 | 26.34 | 29.95 | 33.25 | 35.42 | 37.73 | 39.95 | 43.95 | 1097293305.30 |
| GCN | Z500 ACC | 0.280 | 0.155 | 0.068 | 0.020 | -0.011 | -0.032 | -0.052 | -0.070 | -0.101 |
| | Z500 RMSE | 1897.27 | 2205.72 | 2491.37 | 2616.58 | 2798.23 | 3009.40 | 3272.80 | 3552.62 | 13147.85 |
| | T850 ACC | 0.287 | 0.170 | 0.087 | 0.030 | -0.006 | -0.025 | -0.033 | -0.031 | 0.093 |
| | T850 RMSE | 6.15 | 7.02 | 8.29 | 10.90 | 21.75 | 61.01 | 173.60 | 466.71 | 930146229837458.25 |
| GemCNN | Z500 ACC | 0.934 | 0.984 | 0.962 | 0.910 | 0.818 | 0.742 | 0.650 | 0.584 | -0.007 |
| | Z500 RMSE | 304.11 | 156.00 | 242.86 | 364.36 | 521.75 | 615.11 | 724.60 | 785.92 | 1461.45 |
| | T850 ACC | 0.879 | 0.995 | 0.901 | 0.776 | 0.670 | 0.581 | 0.477 | 0.392 | -0.025 |
| | T850 RMSE | 1.71 | 0.34 | 1.51 | 2.30 | 2.77 | 3.10 | 3.49 | 3.80 | 6.05 |
| Hermes | Z500 ACC | 0.919 | 0.948 | 0.888 | 0.795 | 0.670 | 0.568 | 0.472 | 0.402 | 0.129 |
| | Z500 RMSE | 339.23 | 289.93 | 463.62 | 667.90 | 927.73 | 1157.17 | 1416.43 | 1653.81 | 25644.40 |
| | T850 ACC | 0.842 | 0.906 | 0.756 | 0.586 | 0.447 | 0.340 | 0.252 | 0.187 | -0.025 |
| | T850 RMSE | 1.96 | 1.50 | 2.55 | 3.59 | 4.45 | 5.20 | 5.99 | 6.76 | 293.82 |
| MPNN | Z500 ACC | 0.911 | 0.943 | 0.916 | 0.839 | 0.726 | 0.617 | 0.512 | 0.429 | 0.139 |
| | Z500 RMSE | 349.58 | 283.77 | 360.45 | 519.00 | 721.06 | 908.69 | 1116.36 | 1310.80 | 11191.46 |
| | T850 ACC | 0.846 | 0.933 | 0.898 | 0.817 | 0.724 | 0.638 | 0.547 | 0.469 | 0.043 |
| | T850 RMSE | 1.90 | 1.23 | 1.52 | 2.03 | 2.48 | 2.82 | 3.17 | 3.44 | 5.99 |
| Transformer | Z500 ACC | -0.150 | -0.150 | -0.150 | -0.150 | -0.150 | -0.150 | -0.149 | -0.149 | -0.147 |
| | Z500 RMSE | 55368.55 | 55368.50 | 55368.45 | 55520.22 | 55520.17 | 55520.12 | 55558.25 | 55558.18 | 55570.73 |
| | T850 ACC | 0.884 | 0.884 | 0.884 | 0.553 | 0.553 | 0.553 | 0.304 | 0.304 | -0.068 |
| | T850 RMSE | 1.62 | 1.62 | 1.62 | 3.13 | 3.13 | 3.13 | 3.96 | 3.96 | 7.37 |
| R-UNIMESH (**Ours**) | Z500 ACC | 0.950 | 0.985 | 0.947 | 0.882 | 0.786 | 0.703 | 0.614 | 0.546 | 0.147 |
| | Z500 RMSE | 260.31 | 140.60 | 271.28 | 405.01 | 556.83 | 662.19 | 773.67 | 853.87 | 2821.03 |
| | T850 ACC | 0.888 | 0.964 | 0.889 | 0.774 | 0.679 | 0.598 | 0.504 | 0.428 | 0.134 |
| | T850 RMSE | 1.63 | 0.91 | 1.63 | 2.37 | 2.85 | 3.20 | 3.65 | 4.04 | 11.68 |

*Table 13.* Weather Forecasting Results, ACC and RMSE at Different Lead Times for Temperature and Geopotential.

| Model | RE Temperature ($\downarrow$) | RE Geopotential ($\downarrow$) |
|---|---|---|
| GCN | $6.8 \cdot 10^{-2} \pm 1.7 \cdot 10^{-2}$ | $1.2 \cdot 10^{-3} \pm 1.0 \cdot 10^{-3}$ |
| MPNN | $3.0 \cdot 10^{-3} \pm 2.7 \cdot 10^{-3}$ | $1.3 \cdot 10^{-3} \pm 8.5 \cdot 10^{-4}$ |
| Hermes | $3.1 \cdot 10^{-1} \pm 7.5 \cdot 10^{-1}$ | $1.0 \cdot 10^{-1} \pm 3.6 \cdot 10^{-1}$ |
| GemCNN | $\mathbf{8.9 \cdot 10^{-4} \pm 7.0 \cdot 10^{-4}}$ | $9.4 \cdot 10^{-4} \pm 6.1 \cdot 10^{-4}$ |
| EMAN | $7.7 \cdot 10^{0} \pm 5.2 \cdot 10^{0}$ | $4.1 \pm 2.6 \cdot 10^{-2}$ |
| EGNN | $1.1 \cdot 10^{-3} \pm 7.3 \cdot 10^{-4}$ | $\mathbf{8.3 \cdot 10^{-4} \pm 1.0 \cdot 10^{-3}}$ |
| Transformer | $9.4 \cdot 10^{-4} \pm 8.2 \cdot 10^{-4}$ | $2.0 \cdot 10^{-3} \pm 1.1 \cdot 10^{-3}$ |
| R-UNIMESH (Ours) | $2.2 \cdot 10^{-3} \pm 1.6 \cdot 10^{-3}$ | $9.8 \cdot 10^{-4} \pm 7.2 \cdot 10^{-4}$ |

*Table 14.* Rayleigh error for all models for all initializations on WB2 temperature and geopotential. Best performing model is indicated with **bold text**. Errors scaled up by $\times 40$.

