# OpenReview forum: "Smoothness Errors in Dynamics Models and How to Avoid Them"
_ICML.cc/2026/Conference — ICML 2026 regular_

### Official Review · Reviewer_1z8n · 2026-03-09

**Soundness:** 3
**Presentation:** 4
**Significance:** 3
**Originality:** 3
**Overall Recommendation:** 5
**Confidence:** 3

**Summary:**

This paper studies smoothness control in mesh-based dynamics modeling. Standard message-passing GNNs often oversmooth, while recently proposed unitary convolutions exactly preserve the Rayleigh quotient and therefore avoid oversmoothing. The paper argues, however, that exact smoothness preservation can be overly restrictive for dynamics whose smoothness naturally evolves over time, especially diffusion-like processes. To support this claim, it derives a lower approximation error bound for unitary functions and shows that unitary models become overconstrained when the target function has strong angular dependence in its output norm.

Motivated by this analysis, the paper proposes two relaxations of unitary convolution. The first uses Taylor truncation of the matrix exponential in Lie unitary convolution, yielding an interpretable and controllable relaxation of the strict unitary constraint. The second adopts an encoder-decoder design with a unitary encoder and an unconstrained decoder to improve practicality and scalability. The paper also extends the Rayleigh quotient and unitary convolution from graphs to meshes using cotangent-weighted mesh Laplacians and proves the corresponding invariance results under suitable assumptions.

Experiments are conducted on heat diffusion over grid graphs, several PDEs on triangular meshes, and weather forecasting on an Earth mesh. The results indicate that the relaxed models are particularly effective on diffusive dynamics, where matching target smoothness is especially important, and that the mesh-based variant is competitive with strong mesh-aware baselines.

**Compliance With Llm Reviewing Policy:**

Affirmed.

**Final Justification:**

The paper makes a meaningful and technically solid contribution by identifying a real limitation of strict unitary smoothness preservation and proposing practical relaxations that work well, especially in diffusion-like settings. The rebuttal was helpful in clarifying scope and adding empirical context, although my original reservations about the broader generality and novelty of the approach remain to some extent. Overall, I found the strengths sufficient to support acceptance, and the discussion did not change my overall evaluation, so I am maintaining my original score.

**Key Questions For Authors:**

All key questions are addressed by the points raised in the weaknesses section above.

**Limitations:**

No, the manuscript does not discuss its limitations or potential broader impacts. In particular, the theoretical lower bound depends on the angular dependence of the target function $f$, but this quantity may be difficult to estimate or operationalize in practice. It would therefore be helpful for the paper to discuss more explicitly when the bound can meaningfully guide model selection and when its practical applicability is limited.

**Strengths And Weaknesses:**

## Strengths

- **Clear and well-motivated idea.** The paper addresses the tradeoff between oversmoothing and undersmoothing in GNN-based dynamics modeling. For Laplacian-driven dynamics, this is a natural and important question, and the claim that strict unitary convolution can induce undersmoothing is clearly motivated.

- **Meaningful theoretical contribution.** The lower approximation error bound for unitary functions is a substantive result, as it guides the choice of GNN architecture to the angular dependence of the target function. The extension of the Rayleigh quotient and unitary convolution framework from graphs to meshes further increases the technical and practical relevance of the work.

- **Well-developed methodology.** The paper goes beyond identifying a limitation of prior unitary models and proposes two concrete relaxations. The Taylor-truncation variant is especially clean and interpretable, while the encoder-decoder variant is practically motivated and scalable.

- **Strong presentation and related work.** The manuscript is generally well organized, and the related work section is particularly effective in positioning the paper with respect to oversmoothing, graph and mesh dynamics modeling, and PDE surrogate benchmarks.

## Weaknesses

- **Limited topology coverage.** Although the motivation and theory are stated for graphs more broadly, the experiments are almost entirely restricted to highly structured spatial graphs, such as grid graphs, triangular meshes, and an Earth mesh. It therefore remains unclear whether the proposed ideas extend to non-geometric graphs with more heterogeneous connectivity. Even a simple study on random graphs would have made the empirical claims more convincing.

- **Benefits appear strongest on diffusive dynamics.** The heat equation results are compelling, but they also represent the regime in which Laplacian-based smoothness control is most naturally aligned with the target operator. The gains are less consistent on wave and Cahn-Hilliard dynamics, which weakens the case that the method reflects a broadly useful principle rather than a particularly good match to diffusion-dominated settings.

- **Limited novelty of the practical architecture.** The encoder-decoder relaxation appears close to a fairly standard design in which a unitary encoder is followed by a conventional GNN decoder. This makes it difficult to identify what is specifically novel in the encoder-decoder architecture?

- **Positioning against operator-learning baselines is incomplete.** The evaluated tasks are naturally viewed as PDE surrogate modeling on meshes, yet the comparisons are limited to mesh GNN baselines. Including operator-learning methods such as Graph Neural Operators or MeshGraphNet would have clarified the contribution's position within the broader scientific ML literature.

- **Minor presentation issue.** Fig. 2 may be easier to follow if the Taylor-truncation variant is plotted on the top row, since both the legend and the main text introduce the Taylor truncation variant first.

---

> ### Author Rebuttal · Authors · 2026-03-31
>
> > Limited topology coverage. Although the motivation and theory are stated for graphs more broadly, the experiments are almost entirely restricted to highly structured spatial graphs [...] It therefore remains unclear whether the proposed ideas extend to non-geometric graphs with more heterogeneous connectivity. Even a simple study on random graphs would have made the empirical claims more convincing.
>
> Thank you for this insight! We use grid graphs and triangular meshes in our experiments because we are trying to approximate dynamics over surfaces, and properly discretizing a surface requires structured geometric graphs. While we agree this experiment would improve the paper, we view this as an opportunity for future work.
>
> > Benefits appear strongest on diffusive dynamics. The heat equation results are compelling, but they also represent the regime in which Laplacian-based smoothness control is most naturally aligned with the target operator. The gains are less consistent on wave and Cahn-Hilliard dynamics, which weakens the case that the method reflects a broadly useful principle rather than a particularly good match to diffusion-dominated settings.
>
> While we agree that our method performs particularly well on diffusive dynamics, we note that our method still performs very well compared to strong baselines on the Wave and Cahn-Hilliard datasets. Our method outperforms other models in terms of NRMSE for Wave and is within error bars of the best model for NRMSE and RE for Cahn-Hilliard.
>
> > Limited novelty of the practical architecture. The encoder-decoder relaxation appears close to a fairly standard design in which a unitary encoder is followed by a conventional GNN decoder. This makes it difficult to identify what is specifically novel in the encoder-decoder architecture?
>
> Thanks for your question. Our main source of novelty is our smoothness preserving zero pad procedure, which differs from most prior work on unitary graph convolutions (e.g. [1-2]) which use learned embeddings. This allows for a strict separation of unitary and unitary breaking pieces.  We support the importance of this design choice with an ablation below:
>
> | PDE \ Embedding | Zero Pad     | GCN Mapping      |
> | --------------- | ------------ | ---------------- |
> | Heat NRMSE      | 51.9 ± 3.6   | 4,689.6 ± 232.0  |
> | Heat SMAPE      | 79.7 ± 5.6   | 367.7 ± 0.6      |
> | Heat RE         | 9.1 ± 7.4    | 153.9 ± 13.4     |
> | Wave NRMSE      | 236.5 ± 6.4  | 12532.0 ±  242.2 |
> | Wave SMAPE      | 385.2 ± 1.2  | 383.9 ± 0.6      |
> | Wave RE         | 93.5 ± 25.4  | 11.8 ± 2.8       |
> | CH NRMSE        | 123.9 ± 2.6  | 228.5 ± 1.0      |
> | CH SMAPE        | 167.3 ± 10.6 | 250.8 ± 1.3      |
> | CH RE           | 18.9 ± 10.4  | 9.0 ± 4.3        |
>
> Moreover, while we agree that using a decoder or readout layer is fairly standard practice, we note that we are distinct from most prior work [1-2] in that we test MLPs as decoders rather than just GNNs and found this aided performance on the PDEs task.
>
> [1] Kiani et al. NeurIPS 2024 Spotlight. Unitary Convolutions for Learning on Graphs and Groups
>
> [2] Fesser at al. Unitary Convolutions for Message-Passing and Positional Encodings on Directed Graphs. https://openreview.net/forum?id=xfwlgpe6st
>
> > Positioning against operator-learning baselines is incomplete. The evaluated tasks are naturally viewed as PDE surrogate modeling on meshes, yet the comparisons are limited to mesh GNN baselines. Including operator-learning methods such as Graph Neural Operators or MeshGraphNet would have clarified the contribution's position within the broader scientific ML literature.
>
> Thanks for your suggestion. We have added MeshGraphNet as a baseline. We find that our model outperforms MeshGraphNet on heat and is competitive on Wave and Cahn-Hilliard.
>
> | PDE        | R-UniMesh    | MeshGraphNet |
> | ---------- | ------------ | ------------ |
> | Heat NRMSE | 51.9 ± 3.6   | 108.1 ± 3.2  |
> | Heat SMAPE | 79.7 ± 5.6   | 144.1 ± 6.3  |
> | Heat RE    | 9.1 ± 7.4    | 53.4 ±1.6    |
> | Wave NRMSE | 236.5 ± 6.4  | 253.6 ± 28.2 |
> | Wave SMAPE | 385.2 ± 1.2  | 278.6 ± 2.3  |
> | Wave RE    | 93.5 ± 25.4  | 30.0 ± 23.2  |
> | CH NRMSE   | 123.9 ± 2.6  | 122.0 ± 1.5  |
> | CH SMAPE   | 167.3 ± 10.6 | 186.1 ± 2.9  |
> | CH RE      | 18.9 ± 10.4  | 6.2 ± 0.4    |
>
> > [...] It would therefore be helpful for the paper to discuss more explicitly when the bound can meaningfully guide model selection and when its practical applicability is limited.
>
> This is a great suggestion. The ability to estimate the lower bound depends on (1) the ability to estimate the data distribution and (2) the ability to write the ground truth function in angular coordinates so that the variance of ||f|| over each sphere can be computed. We note that appendix A5 already contains a worked example that demonstrates this. We are happy to discuss this and other limitations more in the main text in a camera ready version.

---

> > ### Author Rebuttal · Reviewer_1z8n · 2026-04-03
> >
> > The rebuttal is helpful and clarifies the intended scope of the paper.
> >
> > However, my overall assessment remains unchanged. The theoretical and empirical case is strongest for diffusive dynamics, and I agree that the paper supports this setting well. For non-diffusive dynamics, however, the rebuttal does not resolve my main concern. The theoretical justification is still limited, and the empirical results remain mixed rather than clearly strong.
> >
> > I also appreciate the additional clarification on the encoder-decoder relaxation, but these do not substantially change my view of novelty. Zero padding seems to be the most straightforward way to preserve unitarity while increasing dimension, and the MLP decoder appears to be a standard shared per-node readout.
> >
> > Overall, I find the direction of relaxed unitary convolution promising, but the remaining concerns about scope and methodological novelty are not resolved. I will therefore keep my recommendation unchanged.

---

> > > ### Author Response · Authors · 2026-04-05
> > >
> > > We appreciate that you find this direction promising, thank you again for your detailed review!

---

### Official Review · Reviewer_iGum · 2026-03-11

**Soundness:** 3
**Presentation:** 4
**Significance:** 3
**Originality:** 4
**Overall Recommendation:** 4
**Confidence:** 3

**Summary:**

The paper proposes relaxed unitary convolutions for GNN-based PDE simulation on meshes. The core insight is that standard unitary convolutions are provably overconstrained for diffusive dynamics. Theorem 1 gives an explicit lower bound on their approximation error tied to how much the target function changes signal norms. Two relaxations are proposed: truncating the Lie convolution Taylor series (R-UNIGRAPH) and an encoder-decoder architecture with a unitary encoder and unconstrained decoder (R-UNIMESH). Both are extended to 3D meshes via the cotangent Laplacian with formal Rayleigh quotient preservation guarantees. R-UNIMESH outperforms baselines on heat diffusion across complex 3D meshes and is competitive on wave and Cahn-Hilliard equations.

**Compliance With Llm Reviewing Policy:**

Affirmed.

**Final Justification:**

I am maintaining a Weak Accept. While the authors have addressed nearly all of my concerns, the issue regarding the omission of boundary conditions remains unresolved. However, I believe the paper’s overall contribution and the strengths identified in the other sections still merit acceptance. I recommend that the authors at least acknowledge this limitation in the final version.

**Key Questions For Authors:**

1. how much of R-UNIMESH's performance comes from the unitary encoder versus the decoder? Replacing the unitary encoder with a standard GCN encoder of equal parameter count would clarify whether the inductive bias is genuinely load-bearing or whether the decoder dominates.
2. R-UNIGRAPH uses Tmax=3 throughout. Is there a principled way to select Tmax without grid search, perhaps tied to the expected energy dissipation rate of the target PDE? A data-driven or physics-informed Tmax schedule could make the method more practical.
3. Does the model take only u(t) as input, or a window of past timesteps u(t-k), ..., u(t)? For the wave equation, a single timestep does not encode velocity, which could partly explain R-UNIMESH's underperformance on wave dynamics. If so, would providing u(t) and u(t-Δt) as joint input close the performance gap against EMAN?
4. The paper claims generalization to unseen meshes, but the train/test split is not clearly described. Are test meshes completely held out during training, or does the model see all 7 meshes during training with only initial conditions held out? The strength of the generalization claim depends critically on this distinction.

**Limitations:**

Yes

**Strengths And Weaknesses:**

Strengths
1. Theorem 1 gives a rigorous approximation bound which is expressed in terms of the variance of ||f|| on concentric spheres, directly quantifying how much unitary functions fail rather than just showing they fail. This gives a principled metric for when relaxation is necessary.
2. Rayleigh quotient used coherently throughout. The same metric diagnoses the problem (Figure 3 showing Lie Uni stuck at input RQ), motivates the theory (Theorem 1), drives the relaxation design (Tmax controls RQ deviation exponentially per Figure 8), and evaluates results (MRE in Table 1).
3. Taylor Truncation provides a precise relaxation dial. Figure 8 shows KL divergence between input and output RQ distributions decays exponentially with Tmax, meaning you can predictably trade off smoothness preservation against modeling flexibility with a single integer hyperparameter.
4. Cotangent Laplacian extension is geometrically principled. Replacing uniform edge weights with cotangent weights means the convolution approximates the true Laplace-Beltrami operator on the surface, not just a topological adjacency. Corollary 1 formally extends the RQ preservation guarantee to this geometry-aware setting under the Delaunay condition.

Weaknesses
1. There exists inductive bias mismatch for non-diffusive PDEs. For example, R-UNIMESH loses to EMAN on SMAPE and RE for wave equation (Table 2). A dynamic Tmax or PDE-type-conditioned relaxation might better adapt smoothness control to oscillatory dynamics.
2. Unlike Taylor Truncation where Tmax precisely controls unitarity deviation, the MLP/GCN decoder provides no mathematical handle on how much unitarity is broken.
3. The closed manifold restriction reduces PDEs to ODEs since all meshes are closed surfaces with no boundary. Real engineering meshes (flat plates, open shells) require boundary conditions. Extending Corollary 1 to handle Dirichlet or Neumann boundaries would significantly broaden applicability.
4. The Cahn-Hilliard results are weak. GemCNN outperforms R-UNIMESH on NRMSE and RE for Cahn-Hilliard (Table 2). Phase separation dynamics involve sharp interfaces rather than global smoothness changes, suggesting the Rayleigh quotient may be too coarse a smoothness measure for interface-dominated dynamics. A localized or multi-scale smoothness metric could better capture these phenomena.
5. No ablation on decoder architecture. It is unclear whether MLP vs GCN decoder choice meaningfully affects results, or how sensitive performance is to decoder depth and width. A systematic ablation would clarify how much work the decoder is actually doing versus the unitary encoder.

---

> ### Author Rebuttal · Authors · 2026-03-31
>
> Thanks for your constructive review!
>
> > There exists inductive bias mismatch for non-diffusive PDEs. [...] A dynamic Tmax or PDE-type-conditioned relaxation might better adapt smoothness control to oscillatory dynamics.
>
> We implemented a version of R-UniMesh with dynamic T_max and found that it was outperformed by our original framework, though careful tuning may improve the viability.
>
> > [...] The MLP/GCN decoder provides no mathematical handle on how much unitarity is broken.
>
> While the encoder-decoder method sacrifices precise control over how much unitarity is broken, it allows varying the hidden dimension without increasing depth for improved scalability. We view the two relaxation strategies as complementary: Taylor truncation offers more fine-grained control when the smoothness budget is known a priori, while the encoder-decoder is preferred when scalability is the priority.
>
> > [...] Real engineering meshes (flat plates, open shells) require boundary conditions. Extending Corollary 1 to handle Dirichlet or Neumann boundaries would significantly broaden applicability.
>
> This is a great suggestion. We will mention this as future work in the conclusion.
>
> > The Cahn-Hilliard results are weak. GemCNN outperforms R-UNIMESH on NRMSE and RE for Cahn-Hilliard (Table 2). Phase separation dynamics involve sharp interfaces rather than global smoothness changes, suggesting the Rayleigh quotient may be too coarse a smoothness measure for interface-dominated dynamics. A localized or multi-scale smoothness metric could better capture these phenomena.
>
> Thanks for your suggestion. We add a localized Rayleigh quotient and show that R-UniMesh is able to capture the local smoothness tendencies on some trajectories, albeit less consistently than GemCNN. In particular, we sample subgraphs from our meshes and study the distribution of Rayleigh quotients of the subgraphs for both the predicted and target signals. On the Cahn-Hilliard task, we found that GemCNN was more consistent in matching the distribution over time, which explains the performance in Table 3.
>
> The Cahn-Hilliard results are [here](https://anonymous.4open.science/r/rayleigh_analysis-BD52/rebuttal/localized_metric/ch_gifs/).
>
> We also compare the Rayleigh quotient against an alternative multiscale smoothness metric, the 2-point correlation function, in Appendix C.5. We find that they are good proxies for each other in the dynamics we study.
>
> > No ablation on decoder architecture. It is unclear whether MLP vs GCN decoder choice meaningfully affects results, or how sensitive performance is to decoder depth and width. [...]
>
> We conduct additional ablations showing that the decoder does some work as illustrated by different decoder heads performing better for different tasks. However, performance is not sensitive to small changes in width and depth. We found that the models diverged on the PDE task when using a GCN decoder and converged using MLPs. However, we observed the opposite behavior for WB2. Depth and width ablations on the MLP decoder for the PDE task are provided below. The results can be found [here](https://anonymous.4open.science/r/rayleigh_analysis-BD52/table.png)
>
> > how much of R-UNIMESH's performance comes from the unitary encoder versus the decoder? Replacing the unitary encoder with a standard GCN encoder of equal parameter count would clarify whether the inductive bias is genuinely load-bearing.
>
> We provide this ablation in our response to reviewer zyET and find that unitary encoders prove to be important for diffusive dynamics and remain competitive for wave and Cahn-Hilliard.
>
> > Is there a principled way to select Tmax without grid search, perhaps tied to the expected energy dissipation rate of the target PDE? [...]
>
> Yes, there is a principled way to find $T_{\rm max}$ . Proposition 6 can be used as a model selection criterion for selecting $T_{\rm max}$ based on the amount of change in the Rayleigh quotient required by the task (i.e. the expected energy dissipation rate). Different $T_{\rm max}$’s yield different truncation errors that can be estimated by Taylor’s theorem, and we can treat these error estimates as the perturbations delta in the proposition to compute the change in smoothness in our function. Great question!
>
> > Does the model take only u(t) as input, or a window of past timesteps u(t-k), ..., u(t)? [...] would providing u(t) and u(t-Δt) as joint input close the performance gap against EMAN [on Wave]?
>
> A window of past timesteps. For all three PDEs, the window size is 5. For weather forecasting, the window size is 3. Therefore, providing a window of timesteps does not close the performance gap.
>
> > [...] Are test meshes completely held out during training, or does the model see all 7 meshes during training with only initial conditions held out? [...]
>
> Test meshes are completely held out during training for PDEs. For Weatherbench all predictions are made on the Earth mesh, so only initial conditions were held out.

---

> > ### Author Rebuttal · Reviewer_iGum · 2026-04-02
> >
> > I thank the authors for their comprehensive responses. While most of my concerns have been addressed, a few key points remain:
> >
> > Weakness 2: The response did not directly clarify how unitarity deviation is measured and controlled for the MLP/GCN decoders.
> >
> > Weakness 3: This is an important issue however i don't think it can be easily addressed without significant update to the paper.
> >
> > Given these remaining points, I will maintain my original score.

---

> > > ### Author Response · Authors · 2026-04-05
> > >
> > > Thank you again for the very comprehensive review of the paper, we appreciate the time and effort taken to give constructive and thoughtful feedback to our work!
> > >
> > > > Weakness 2: The response did not directly clarify how unitarity deviation is measured and controlled for the MLP/GCN decoders.
> > >
> > > Thank you for clarifying, this is a great point. For GCNs, there are tools we can use to measure and control unitary deviation directly. In particular, we can consider GCNs that are initialized with an orthogonal / unitary weight matrix, which was shown to be an effective strategy in [1]. In particular, $f_{\rm GCN} = AXU$, $UU^\dagger = I$ at initialization. Next, [2] notes that separable unitary convolution recovers a GCN layer with unitary weights up to first order. Therefore, we can estimate the unitary deviation at initialization in a GCN by considering the equivalent formulation as a 1-term separable unitary network. We can then apply Taylor’s theorem to the 1-term unitary network to compute the unitary deviation in the GCN.  Viewed in this sense, the decoder already has some measure of control on the deviation from unitarity at initialization. Unfortunately, GCN layers that map to the target node feature dimension remain unaddressed, since unitary weight matrices are not possible in this setting due to dimension mismatch. We did not use any extra regularization or added losses to try and control the extent of unitary breaking in the GCN decoders. We will explore these further in future work.
> > >
> > > For MLPs, measuring and controlling unitary deviation is less straightforward. The MLPs are fully unconstrained in this study. However, future work will explore using unitary layers for the MLP linear layers coupled with auxiliary losses.
> > >
> > > [1] Xiao et al. ICML 2018. Dynamical isometry and a mean field theory of cnns: How to train 10,000-layer vanilla convolutional neural network
> > >
> > > [2] Kiani et al. NeurIPS 2024 Spotlight. Unitary Convolutions for Learning on Graphs and Groups
> > >
> > > > Weakness 3: This is an important issue however i don't think it can be easily addressed without significant update to the paper.
> > >
> > > We agree that extending corollary 1 to different boundary conditions would strengthen the applicability of our paper and will pursue it in future work.

---

### Official Review · Reviewer_vG9x · 2026-03-11

**Soundness:** 3
**Presentation:** 2
**Significance:** 2
**Originality:** 3
**Overall Recommendation:** 3
**Confidence:** 3

**Summary:**

This paper claims that unitary convolutions in GNNs for dynamic systems can be overly restrictive and may limit model performance. To solve this problem, the authors propose a relaxed unitary convolutions design that seeks a better balance between smoothness preservation and performance. Experiments on heat diffusion, mesh dynamics, and weather forecasting show that the proposed method performs competitively or better than strong baselines.

**Compliance With Llm Reviewing Policy:**

Affirmed.

**Final Justification:**

I think the paper still remains unclear on the validation of the bottleneck claim, and there is room to improve the overall presentation and empirical justification. So I would like to maintain my score.

**Key Questions For Authors:**

1. The paper claims that strict unitary structure is bottleneck. Can the authors provide an experiment that more directly validates this claim, rather than only showing downstream performance?

2. How much of the improvement comes specifically from the proposed relaxation mechanism, rather than from increased flexibility or model capacity more generally? A stricter ablation study would help clarify this.

3. The paper discusses both oversmoothing and undersmoothing. Can the authors explain more clearly how these two phenomena are captured in one unified framework, instead of just presenting them as empirical results?

4. How sensitive is the method to the degree of relaxation? Is the performance stable across different settings, or does it require careful tuning?

5. How does the method perform on dynamics with weaker spatial smoothness, such as systems with sharp interfaces or discontinuities?

**Limitations:**

Authors not discuss limitations. A limitation is that the method and analysis rely on the Rayleigh quotient formulation, and it is unclear how well the approach generalizes beyond this setting

**Strengths And Weaknesses:**

# Strengths
1. The core idea is well motivated. The paper claims that unitary convolutions may be too restrictive, and a relaxed design can provide a better tradeoff between smoothness preservation and model expressiveness in learning dynamical systems.

2. Empirical result is generally supportive.
Across several tasks, the relaxed variants appear stronger or more robust than stricter unitary alternatives, which supports the paper’s main claim that some degree of relaxation can be beneficial.

# Weaknesses

1. The main claim is broader than what is fully validated.
The paper presents the problem in general terms, as if it were addressing smoothness errors in GNNs more broadly. However, the theoretical development is tied to a more specific unitary / Rayleigh quotient setting, and the experiments are concentrated on relatively smooth benchmarks such as heat diffusion and weather forecasting. This makes the scope of the claims somewhat broader than the actual empirical support.

2. The mechanism is not fully verified by the experiments.
The results show a correlation between better smoothness behavior and better predictive performance, but not provide strong evidence that the proposed relaxation mechanism is the main reason for the improvement. It remains possible that part of the gains comes from increased flexibility or implementation details rather than the specific principle emphasized in the paper.

3. Boundary conditions of applicability are under-discussed.
The paper mainly emphasizes smoothness preservation, but does not sufficiently discuss cases where the target dynamics contain sharp transitions, discontinuities, or multi-scale high-frequency structure. In such settings, smoothness may not always be the dominant prior.

---

> ### Author Rebuttal · Authors · 2026-03-31
>
> > [...] The paper presents the problem in general terms, as if it were addressing smoothness errors in GNNs more broadly. However, the theoretical development is tied to a more specific unitary / Rayleigh quotient setting... This makes the scope of the claims somewhat broader than the actual empirical support.
>
> > Authors not discuss limitations [...]
>
> We believe the Rayleigh quotient is a well-motivated choice for studying smoothness in GNNs for dynamics, as it is a standard measure for smoothness in the literature, see [1-2]. The unitary framework follows naturally as it preserves the Rayleigh quotient. We further support our choice of the Rayleigh quotient by comparing it with an alternative smoothness metric in Appendix C.5 and show that they support the same conclusions.
>
> [1] Kiani et al. Unitary Convolutions for Learning on Graphs and Groups. NeurIPS 2024
>
> [2] Rusch et al. A survey on oversmoothing in graph neural networks. arXiv
>
> > experiments are concentrated on relatively smooth benchmarks [...]
>
> > How does the method perform on dynamics with weaker spatial smoothness [...]
>
> We note that although heat diffusion and global weather forecasting are indeed relatively smooth, the Cahn-Hilliard equation describes phase separation dynamics that contain sharp discontinuities. This is also pointed out by reviewer iGum.
>
> > [...] The results show a correlation between better smoothness behavior and better predictive performance, but not provide strong evidence that the proposed relaxation mechanism is the main reason for the improvement. [...]
>
> > [...] A stricter ablation study would help clarify this.
>
> We assume the mechanism refers to the encoder-decoder method. We performed additional ablations to isolate the contribution of the relaxation mechanism. Our results validate the design of our method and confirm that the relaxation mechanism is critical, especially for diffusive dynamics. Specifically, our ablation study supports the claim that the encoder should preserve smoothness at least approximately and in many cases strictly. Our method can still perform well even when subtle smoothness changing modifications are made to the encoder. This includes early Taylor truncation in the unitary encoder and ReLU activation functions. However, the smoothness preserving zero padding procedure is crucial to performance.
>
> Our results table for our ablations are in our response to reviewer zyET.
>
> > The paper [...] does not sufficiently discuss cases where the target dynamics contain sharp transitions, discontinuities, or multi-scale high-frequency structure. In such settings, smoothness may not always be the dominant prior.
>
> We agree that the selection of suitable prior is important, and smoothness may not always be the dominant prior. We will include a more nuanced discussion of limitations in the revision.
>
> > The paper claims that strict unitary structure is bottleneck. Can the authors provide an experiment that more directly validates this claim, rather than only showing downstream performance?
>
> Figure 3 gives direct evidence  that unitary structure is a bottleneck by overconstraining the model to preserve smoothness when the underlying task changes smoothness, which agrees with Proposition 1 and Theorem 1 in our paper. This in turn generally hurts downstream performance in terms of MSE as shown in Table 1.
>
> > The paper discusses both oversmoothing and undersmoothing. Can the authors explain more clearly how these two phenomena are captured in one unified framework [...]?
>
> Thank you for bringing this up. We define both oversmoothing and undersmoothing of a model f with respect to the ground truth y. If predicted dynamics are smoother than the ground truth (R(f(X)) < R(y)), then f is oversmoothing. If predicted dynamics are less smooth than the ground truth (R(f(X)) > R(y)), then we say that the model is undersmoothing. We are happy to provide formal definition blocks in the revision.
>
> > How sensitive is the method to the degree of relaxation? Is the performance stable across different settings, or does it require careful tuning?
>
> We find that the Taylor-truncation method is relatively stable across different truncation levels. We quantify sensitivity to $T_{\rm max}$ with new experiments [here](https://anonymous.4open.science/r/rayleigh_analysis-BD52/rebuttal/truncation/results/)
>
> While $T_{\rm max}$=3 best satisfies both Rayleigh and MSE losses, our method can achieve low MSE loss for $T_{\rm max}$ values between 1 and 5, suggesting that our method can perform well under a range of $T_{\rm max}$ values without careful tuning.
>
> We also point to Proposition 6, which can be used as a model selection criterion for selecting $T_{\rm max}$ based on the task. Different $T_{\rm max}$’s yield different truncation errors that can be estimated by Taylor’s theorem, and we can treat these error estimates as the perturbations $\delta$ in the proposition to compute the change in smoothness in our function.

---

> > ### Author Rebuttal · Reviewer_vG9x · 2026-04-02
> >
> > Thanks for the authors' rebuttal. I think several concerns remain unresolved.
> >
> > 1. I am not convinced by the authors' response on the scope of the claims. My concern was not whether the Rayleigh quotient is a reasonable smoothness metric, but whether the paper’s claims are broader than what is actually validated.
> >
> > 2. The ablations appear to show that some design choices are important, but they do not isolate whether the gains come specifically from the proposed relaxation principle, rather than from increased flexibility or implementation details.
> >
> > 3. Regarding the claim that strict unitary structure is bottleneck. My request was for a more direct validation of the bottleneck claim, beyond downstream performance trends. The rebuttal mainly points back to Figure 3 and the existing theory, which is not a sufficiently direct validation.

---

> > > ### Author Response · Authors · 2026-04-05
> > >
> > > > I am not convinced by the authors' response on the scope of the claims. My concern was not whether the Rayleigh quotient is a reasonable smoothness metric, but whether the paper’s claims are broader than what is actually validated.
> > >
> > > Thank you for clarifying your comment. Our main goal was not to claim that we address smoothness errors in GNNs in full generality. Instead, our goal was to state that while unitary convolutions are useful, they are overconstraining for many dynamical systems by preserving the Rayleigh quotient. We are happy to update the manuscript to better reflect this.
> > >
> > > > The ablations appear to show that some design choices are important, but they do not isolate whether the gains come specifically from the proposed relaxation principle, rather than from increased flexibility or implementation details.
> > >
> > > We respectfully disagree that our ablations do not illustrate that the gains come specifically from the proposed relaxation principle. We also note that in addition to the set of ablations we referred to in our initial rebuttal, we also provided a set of width / depth decoder ablations for reviewer iGUM, showing that our method was not overly sensitive to these changes. These ablations are below:
> > >
> > > [Decoder Ablations](https://anonymous.4open.science/r/rayleigh_analysis-BD52/table.png).
> > >
> > > We also point out that both reviewers zyET and iGUM identified the encoder as the core ablation that would best reflect the claim that our proposed relaxation mechanism was responsible for performance, which we provided in our initial rebuttal. Finally, we also considered a purely unitary model with no decoder as an ablation, however, as we note in the text this would force the network to have exceptionally large depth in order to meet the parameter threshold we set in our experiments. This is because unitary layers are constrained to preserve width. Moreover, this approach would be impossible to implement in settings where many previous timesteps are provided in the input node features, resulting in a dimension mismatch between input and target node features. Including multiple previous timesteps is very common in the literature, including on the benchmarks we use, see [1,2]. For these reasons, we felt that it would be unfair to use a purely unitary model with no decoder as a baseline and found our encoder ablation to be the most appropriate substitute, but we are happy to include this in a revision as well.
> > >
> > > While we have isolated the effect of the proposed relaxation mechanism to the best of our ability, we are happy to add more experiments in the revision if asked of us. Thanks for your comment and continued engagement!
> > >
> > > [1] Park et al. NeurIPS 2023 Modeling Dynamics over Meshes with Gauge Equivariant Nonlinear Message Passing
> > >
> > > [2] Rasp et al. Journal of Advances in Modeling Earth Systems. Weatherbench 2: A benchmark for the next generation of data-driven global weather models.
> > >
> > > > Regarding the claim that strict unitary structure is bottleneck. My request was for a more direct validation of the bottleneck claim, beyond downstream performance trends. The rebuttal mainly points back to Figure 3 and the existing theory, which is not a sufficiently direct validation.
> > >
> > > We apologize if we misunderstood your concern. Our claim is that the unitary structure bottlenecks the model’s ability to adapt smoothness to the underlying task, which we feel was directly validated by Figure 3 and the existing theory in the original text. If there is a specific experiment or theoretical contribution that you would like us to make we are happy to attempt it for revision.

---

### Official Review · Reviewer_zyET · 2026-03-13

**Soundness:** 3
**Presentation:** 3
**Significance:** 2
**Originality:** 3
**Overall Recommendation:** 4
**Confidence:** 4

**Summary:**

The paper studies influence of unitary convolutions on smoothness preservation for dynamics modeling on graphs and meshes. The authors claim that fully unitary convolutions are too restrictive. To address this, the authors derive a lower bound for unitary approximation error and propose relaxed unitary convolutions. Also, they extend the Rayleigh-quotient view from graphs to meshes, and evaluate the resulting models on heat, wave, and Cahn–Hilliard equations as well as reduced-scale WeatherBench2 forecasting.

**Compliance With Llm Reviewing Policy:**

Affirmed.

**Final Justification:**

I think that the paper still has much room for improvement and the broader applicability of the proposed relaxed unitary convolutions is still not clear, so my grade increases to weak acceptance (4).

**Key Questions For Authors:**

1.	Can the authors provide a more controlled comparison in which model capacity and decoder design are fixed, and only strict versus relaxed unitarity is changed?
2.	The theory points to target functions with strong angular dependence of the norm. Can the authors measure or approximate this quantity on the benchmark tasks?
3.	Why does the method help so clearly on heat, but much less consistently on wave and Cahn–Hilliard?

**Limitations:**

The limitations discussion is not sufficient. The paper should state more explicitly that the strongest gains are on diffusive dynamics, that the weather study is reduced-scale rather than fully competitive, and that the empirical gains are not fully disentangled from the architectural choices.

**Strengths And Weaknesses:**

Strengths. The paper addresses an important question, when smoothness-preserving inductive biases become counterproductive for physical dynamics. The perspective is interesting, and the extension from graphs to meshes is practically useful. The heat-diffusion evidence is convincing: in the motivating graph experiment, the relaxed model improves both MSE and Rayleigh error over GCN and Lie-unitary baselines, and on the mesh heat task it shows best results on all reported metrics.

Weaknesses. The central empirical claim is broader than the evidence. The method is clearly strongest on diffusive dynamics, especially heat, but the gains are much less convincing on wave and Cahn–Hilliard.
The practical model is not just “relaxed unitarity”: it also uses an encoder-decoder construction, zero padding, GroupSort, orthogonal weights, and an unconstrained decoder. Since the decoder is exactly the component that breaks the unitary constraint, it is difficult to tell how much of the gain should be credited to the proposed relaxation principle itself, rather than to added flexibility from the surrounding architecture. The ablations do not fully isolate this point.
Third, the theory is interesting but only partially connected to the final experiments. The lower bound shows that unitary functions are overconstrained when the target norm has strong angular dependence, but the paper does not really measure this quantity on the benchmark tasks or demonstrate that it explains the observed task differences.
Finally, the WeatherBench2 experiment is useful but limited. The paper uses only two variables and a reduced training setup due to compute constraints, and it explicitly states that all models remain below benchmark SOTA. That is acceptable for an exploratory study, but it weakens any strong practical claim about weather forecasting performance.

---

> ### Author Rebuttal · Authors · 2026-03-31
>
> > [...] The method is clearly strongest on diffusive dynamics, especially heat, but the gains are much less convincing on wave and Cahn–Hilliard.
>
> We agree and will temper the claims in the revision to reflect the empirical evidence, and clarify our limitations on the Wave and Cahn-Hilliard datasets.
>
> > The practical model is not just “relaxed unitarity”: [...] it is difficult to tell how much of the gain should be credited to the proposed relaxation principle [...]
>
> > Can the authors provide a more controlled comparison in which model capacity and decoder design are fixed, and only strict versus relaxed unitarity is changed?
>
> Thank you for bringing this up. We conducted additional ablations to isolate individual components. The results validate the design of our method and show that the relaxation mechanism is critical, particularly for diffusive dynamics. Specifically, our ablation study supports the claim that the encoder should preserve smoothness at least approximately and in many cases strictly, but that unconstrained encoders struggle for diffusive dynamics. As seen below, our method can still perform well even when subtle smoothness changing modifications are made to the encoder. This includes early Taylor truncation in the unitary encoder and ReLU activation functions. However, the model fails on heat diffusion when using an unconstrained GCN encoder. We also find that the smoothness preserving zero padding procedure is crucial to performance.
>
> |            | Unitary (T=10) | Unitary (T=3) | GCN          |
> | ---------- | -------------- | ------------- | ------------ |
> | Heat NRMSE | 51.9 ± 3.6     | 70.1 ± 6.3    | 416.9 ± 14.3 |
> | Heat SMAPE | 79.7 ± 5.6     | 76.7 ± 10.4   | 248.8 ± 6.6  |
> | Heat RE    | 9.1 ± 7.4      | 9.0 ± 0.6     | 31.5 ± 8.6   |
> | Wave NRMSE | 236.5 ± 6.4    | 242.2 ± 4.1   | 196.0 ± 0.0  |
> | Wave SMAPE | 385.2 ± 1.2    | 299.1 ± 5.9   | 363.1 ± 3.1  |
> | Wave RE    | 93.5 ± 25.4    | 37.7 ± 4.8    | 193.6 ± 4.3  |
> | CH NRMSE   | 123.9 ± 2.6    | 130.1 ± 2.2   | 175.0 ± 0.9  |
> | CH SMAPE   | 167.3 ± 10.6   | 155.9 ± 7.2   | 187.0 ± 2.8  |
> | CH RE      | 18.9 ± 10.4    | 24.2 ± 8.5    | 12.0 ± 4.6   |
>
> |   | GroupSort    | ReLU         |
> | ---------- | ------------ | ------------ |
> | Heat NRMSE | 51.9 ± 3.6   | 87.7 ± 10.5  |
> | Heat SMAPE | 79.7 ± 5.6   | 82.9 ± 3.6   |
> | Heat RE    | 9.1 ± 7.4    | 9.0 ± 10.5   |
> | Wave NRMSE | 236.5 ± 6.4  | 825.7 ± 12.3 |
> | Wave SMAPE | 385.2 ± 1.2  | 340.2 ± 4.3  |
> | Wave RE    | 93.5 ± 25.4  | 22.4 ± 4.0   |
> | CH NRMSE   | 123.9 ± 2.6  | 144.7 ± 4.2  |
> | CH SMAPE   | 167.3 ± 10.6 | 151.2 ± 4.5  |
> | CH RE      | 18.9 ± 10.4  | 30.0 ± 8.0   |
>
> | | Zero Pad     | GCN Mapping      |
> | --------------- | ------------ | ---------------- |
> | Heat NRMSE      | 51.9 ± 3.6   | 4,689.6 ± 232.0  |
> | Heat SMAPE      | 79.7 ± 5.6   | 367.7 ± 0.6      |
> | Heat RE         | 9.1 ± 7.4    | 153.9 ± 13.4     |
> | Wave NRMSE      | 236.5 ± 6.4  | 12532.0 ±  242.2 |
> | Wave SMAPE      | 385.2 ± 1.2  | 383.9 ± 0.6      |
> | Wave RE         | 93.5 ± 25.4  | 11.8 ± 2.8       |
> | CH NRMSE        | 123.9 ± 2.6  | 228.5 ± 1.0      |
> | CH SMAPE        | 167.3 ± 10.6 | 250.8 ± 1.3      |
> | CH RE           | 18.9 ± 10.4  | 9.0 ± 4.3        |
>
> > [...] the paper does not really measure [the lower bound] on the benchmark tasks [...]
>
> This is a fair point. Unfortunately, the bound can not be computed on the benchmark tasks, as computing it requires (1) the ability to estimate the data distribution and (2) the ability to write the ground truth function in angular coordinates so that the variance of ||f|| over each sphere can be computed. While we can not compute the bound on the tasks for these reasons, we note that Appendix A5 already contains a worked example that demonstrates when and how a bound may be computed.
>
> > WeatherBench2 experiment is useful but limited. [...]
>
> We agree that our reduced training set up is not suitable for making practical claims about weather forecasting., The primary purpose of the experiments was to 1) apply our method to real world data 2) study the tendencies of cross mesh generalization and inductive bias selection of the baseline models.
>
> > Why does the method help  much less consistently on wave and Cahn–Hilliard?
>
> The method helps most clearly on heat because it's the dynamical system where global smoothness measures are most descriptive of the underlying dynamics. For the wave equation, our model is less consistent because the Rayleigh quotient first increases as a function of time before experiencing damped oscillation. In this setting, the biases to preserve smoothness may be overly restrictive and the model is more heavily reliant on the unconstrained decoder to approximate the true dynamics. For Cahn-Hilliard, the PDE models phase separation dynamics where there are sharp discontinuities not fully captured by global smoothness metrics. We point out that our method is still quite strong for both wave and Cahn-Hilliard.

---

> > ### Author Rebuttal · Reviewer_zyET · 2026-04-04
> >
> > I thank the authors for the additional experiments and clarifications. My concerns are mostly resolved. I think that the paper still has much room for improvement and the broader applicability of the proposed relaxed unitary convolutions is still not clear, so my grade increases to weak acceptance (4).

---

> > > ### Author Response · Authors · 2026-04-05
> > >
> > > We are happy that we were able to address most of your concerns in the rebuttal and that you considered this in your grade! Thanks again you for your thorough review!

---

### Decision · Program_Chairs · 2026-04-30

**Decision:**

Accept (regular)

**Comment:**

This paper studies the role of smoothness-preserving inductive biases in dynamics modeling on graphs and meshes, and argues that strictly unitary convolutions can be overly restrictive when the target dynamics naturally change smoothness over time. The paper’s main contribution is a principled relaxation of this constraint, supported by a useful theoretical perspective based on the Rayleigh quotient and by extensions from graphs to meshes. I found the central idea well motivated and technically meaningful, especially for diffusion-like dynamics where exact smoothness preservation is indeed too rigid. The reviewers generally agreed that this is an interesting and relevant question, and several highlighted the value of the theoretical analysis, the mesh extension, and the strong empirical results on heat diffusion. The rebuttal also helped narrow the scope of the claims appropriately and added useful controlled ablations clarifying the importance of the encoder design and the proposed relaxation mechanism.

At the same time, the paper does have limitations. The empirical case is strongest for diffusive dynamics, while the gains for wave and Cahn–Hilliard are more mixed, and the practical encoder-decoder variant does not admit the same clean control of unitarity deviation as the Taylor-truncation relaxation. However, in my judgment, these limitations are well within the range of what can be acknowledged and clarified in revision, and they do not outweigh the paper’s core contribution.

Overall, after considering the paper, rebuttal, and reviewer discussion, I recommend acceptance: the paper offers a clear conceptual contribution, a solid theoretical and empirical foundation for its main claims.